# Landslide Susceptibility Mapping: Analysis of Different Feature Selection Techniques with Artificial Neural Network Tuned by Bayesian and Metaheuristic Algorithms

Farkhanda Abbas [1,*], Feng Zhang [1], Fazila Abbas [2], Muhammad Ismail [3], Javed Iqbal [4], Dostdar Hussain [3], Garee Khan [5], Abdulwahed Fahad Alrefaei [6] and Mohammed Fahad Albeshr [6]

1 School of Computer Science, China University of Geosciences, Wuhan 430074, China; fengzhang@cug.edu.cn
2 Institute of Soil and Environmental Sciences, University of Agriculture Faisalabad, Faisalabad 38000, Pakistan; fazilaabbas802@uaf.edu.pk
3 Department of Computer Science, Karakoram International University, Gilgit 15100, Pakistan; muhammad.ismail@kiu.edu.pk (M.I.); dostdar.hussain@kiu.edu.pk (D.H.)
4 School of Environmental Studies, China University of Geosciences, Wuhan 430074, China; javediqbal@cug.edu.cn
5 School of Geography, Karakoram International University, Gilgit 15100, Pakistan; garee.khan@kiu.edu.pk
6 Department of Zoology, College of Science, King Saud University, P.O. Box 2455, Riyadh 11451, Saudi Arabia; afrefaei@ksu.edu.sa (A.F.A.); albeshr@ksu.edu.sa (M.F.A.)
* Correspondence: shamin0427@cug.edu.cn

**Abstract:** The most frequent and noticeable natural calamity in the Karakoram region is landslides. Extreme landslides have occurred frequently along Karakoram Highway, particularly during monsoons, causing a major loss of life and property. Therefore, it is necessary to look for a solution to increase growth and vigilance in order to lessen losses related to landslides caused by natural disasters. By utilizing contemporary technologies, an early warning system might be developed. Artificial neural networks (ANNs) are widely used nowadays across many industries. This paper's major goal is to provide new integrative models for assessing landslide susceptibility in a prone area in the north of Pakistan. To achieve this, the training of an artificial neural network (ANN) was supervised using metaheuristic and Bayesian techniques: Particle Swarm Optimization (PSO) algorithm, Genetic algorithm (GA), Bayesian Optimization Gaussian Process (BO_GP), and Bayesian Optimization Tree-structured Parzen Estimator (BO_TPE). In total, 304 previous landslides and the eight most prevalent conditioning elements were combined to form a geospatial database. The models were hyperparameter optimized, and the best ones were employed to generate susceptibility maps. The obtained area under the curve (AUC) accuracy index demonstrated that the maps produced by both Bayesian and metaheuristic algorithms are highly accurate. The effectiveness and efficiency of applying ANNs for landslide mapping, susceptibility analysis, and forecasting were studied in this research, and it was observed from experimentation that the performance differences for GA, BO_GP, and PSO compared to BO_TPE were relatively small, ranging from 0.32% to 1.84%. This suggests that these techniques achieved comparable performance to BO_TPE in terms of AUC. However, it is important to note that the significance of these differences can vary depending on the specific context and requirements of the ML task. Additionally, in this study, we explore eight feature selection algorithms to determine the geospatial variable importance for landslide susceptibility mapping along the Karakoram Highway (KKH). The algorithms considered include Information Gain, Variance Inflation Factor, OneR Classifier, Subset Evaluators, principal components, Relief Attribute Evaluator, correlation, and Symmetrical Uncertainty. These algorithms enable us to evaluate the relevance and significance of different geospatial variables in predicting landslide susceptibility. By applying these feature selection algorithms, we aim to identify the most influential geospatial variables that contribute to landslide occurrences along the KKH. The algorithms encompass a diverse range of techniques, such as measuring entropy reduction, accounting for attribute bias, generating single rules, evaluating feature subsets, reducing dimensionality, and assessing correlation and information sharing. The findings of this study will provide valuable insights into the critical geospatial variables associated with landslide susceptibility along the KKH. These insights can aid in the development of

effective landslide mitigation strategies, infrastructure planning, and targeted hazard management efforts. Additionally, the study contributes to the field of geospatial analysis by showcasing the applicability and effectiveness of various feature selection algorithms in the context of landslide susceptibility mapping.

**Keywords:** artificial neural networks; Bayesian techniques; metaheuristic techniques; hyperparameters; feature selection techniques; land sliding

## 1. Introduction

Artificial neural networks are utilized to solve a range of issues in industries, like banking, manufacturing, electronics, and medicine, among others [1]. The backpropagation approach is typically used to train neural networks [2,3]. The backpropagation method can be trained using a variety of algorithms, including gradient descent, conjugate gradient descent, robust, BFGS quasi-Newton, one-step secant, Levenberg–Marquardt, and Bayesian regularization [4]. Some characteristics, such as the number of hidden layers and the number of neurons in each layer, must be defined during the design and training of an ANN; these aspects vary depending on the particular application. There is not a standard, clear approach for selecting these settings. It is utilized in a way that is almost like trial and error, but it requires more computing time and is not very accurate. Therefore, it is required to propose a method to choose the ideal set of parameters that will have the greatest impact on ANN performance.

From the literature, it can be inferred that metaheuristic optimization algorithms have effectively been employed to overcome the computing limitations of traditional landslide predictive models, hence enhancing their performance. The local minimum trap could become a barrier to precise estimations when it comes to ANNs. But recent research has demonstrated that by using metaheuristic techniques, such computational problems can be resolved [5]. Testing novel metaheuristic methods is an important step toward discovering more efficient models, but, as well-known optimizers (such as the PSO and GA) have been sufficiently evaluated. This study's main objective is to assess the effectiveness of four optimization strategies—Particle Swarm Optimization (PSO), Genetic algorithm (GA), Bayesian Optimization Gaussian Process (BO-GP), Bayesian Optimization Tree-structured Parzen Estimator (BO-TPE) —in combination with artificial neural networks (ANN) to produce a map of the KKH region's susceptibility to landslides in northern areas of Pakistan [6].

In the context of artificial neural networks (ANNs), optimization algorithms, such as PSO, GA, and Bayesian Optimization, have been widely used to improve the performance of ANNs in various classification tasks [7]. The following are some specific examples of their applications in the context of ANNs for data classification:

Particle Swarm Optimization (PSO) for ANN

1. PSO has been used to optimize the weights and biases of the ANN to improve its classification accuracy. By searching the weight space, PSO helps to find the optimal set of weights that minimize the prediction error and maximize the classification performance.
2. PSO has been applied to optimize the architecture of the ANN, including the number of hidden layers and the number of neurons in each layer. It helps to determine the optimal network structure that suits the complexity of the classification problem.

Genetic Algorithm (GA) for ANN

1. GA has been used to train ANNs by adjusting the weights and biases of the network. By applying genetic operators, such as crossover and mutation, GA explores the search space of weight configurations to find the best set of weights that lead to improved classification accuracy.

2.    GA has been employed to optimize the architecture of the ANN, including the number of hidden layers and neurons. By evolving populations of networks with different architectures, GA helps to identify the optimal network structure that achieves better classification performance.

Bayesian Optimization for ANN

1.    Bayesian optimization has been used to optimize the hyperparameters of ANNs, such as learning rate, regularization strength, and network architecture. By leveraging Bayesian inference, it explores the hyperparameter space to find the optimal configuration that maximizes the classification accuracy.

2.    Bayesian optimization has been applied to tune the activation functions, dropout rates, and other architectural choices of ANNs. It helps to identify the best combinations of these hyperparameters that result in improved classification performance.

In all these cases, the optimization algorithms (PSO, GA, and Bayesian optimization) play a crucial role in training and fine-tuning ANNs for data classification. They help to search for optimal weights, biases, and hyperparameter configurations, allowing ANNs to achieve higher accuracy and better generalization in classification tasks [8]. By tuning the weights and biases, these algorithms supervise the performance of the ANN in predicting the landslide susceptibility index (LSI). In the context of predicting landslide susceptibility using artificial neural networks (ANNs), the weights and biases of the network are adjusted or tuned by the optimization algorithms (such as Genetic Algorithms, Particle Swarm Optimization, or Bayesian Optimization) to improve the performance of the ANN in predicting the landslide susceptibility index (LSI). The weights and biases in an ANN determine the strength and influence of the connections between neurons in different layers. By adjusting these parameters, the ANN can learn and adapt to the patterns and relationships present in the input data, which in this case are the conditioning elements used to assess landslide susceptibility. During the training phase, the optimization algorithms iterate through different combinations of weights and biases, evaluating the performance of the ANN based on a defined metric (e.g., accuracy and area under the curve). The algorithms aim to find the set of weights and biases that minimize the prediction error or maximize the performance metric, ultimately leading to a more accurate prediction of the LSI. The optimization algorithms use various strategies, such as gradient descent, genetic operators (crossover and mutation), or probabilistic modeling, to explore the search space of weight and bias combinations [9–13]. Through iterations and feedback from evaluating the ANN's predictions, the algorithms progressively adjust the weights and biases to converge towards an optimal solution that yields the best performance in predicting the LSI. By continuously fine-tuning the weights and biases, these optimization algorithms guide the ANN to learn and generalize patterns from the training data, enabling the model to make more accurate predictions of landslide susceptibility based on the conditioning elements. This iterative optimization process helps to improve the effectiveness of the ANN in capturing the complex relationships between the input features and the LSI, leading to more reliable and accurate predictions of landslide susceptibility.

In addition to discussing the fine-tuning of ANNs using metaheuristic and Bayesian algorithms, this study also addresses the challenge of identifying the geospatial variable importance for landslide susceptibility mapping. To achieve this, we employ eight different feature selection algorithms: Information Gain, Variance Inflation Factor (VIF), OneR Classifier, Subset Evaluators, principal components, Relief Attribute Evaluator, correlation, and Symmetrical Uncertainty. Each algorithm brings a unique approach to assessing the relevance and significance of geospatial variables in predicting landslide occurrences. The selection of these feature selection algorithms offers a comprehensive evaluation of different approaches to geospatial variable importance assessment. By leveraging these algorithms, we aim to uncover the most influential variables that contribute to landslide susceptibility mapping in our case study. The outcomes of this research will provide valuable insights for hazard management, infrastructure planning, and proactive decision making to mitigate

the risks associated with landslides along this critical transportation corridor. This study not only contributes to the specific context of landslide susceptibility mapping along the KKH, but also has broader implications for geospatial analysis and hazard management. The findings will enhance our understanding of the geospatial variables influencing landslide occurrences and serve as a foundation for future research and the development of effective strategies to reduce landslide risks in mountainous regions. Analyzing feature selection algorithms helps in identifying the most important geospatial variables for landslide susceptibility mapping. This deepens our understanding of the specific variables that contribute to landslide occurrences along the KKH in Pakistan. By uncovering the key variables, we can gain valuable insights into the underlying mechanisms and processes that influence landslide susceptibility. By determining the geospatial variable importance, we can refine landslide prediction models and improve the accuracy of landslide susceptibility maps. Focusing on the most relevant variables allows us to develop more precise and reliable mapping methodologies. This, in turn, enables better hazard assessment, infrastructure planning, and land management decisions along the KKH.

Analyzing geospatial variable importance helps in proactive hazard management along the KKH. With a better understanding of the critical variables, authorities and stakeholders can implement targeted measures to minimize the risks associated with landslides. This includes implementing early warning systems, designing effective slope stabilization measures, and adopting appropriate land use planning strategies. By identifying the most influential geospatial variables, analyzing feature selection algorithms enables optimal resource allocation for landslide mitigation efforts. Instead of allocating resources uniformly across all variables, decision makers can prioritize and allocate resources to monitor, manage, and mitigate the variables that have the greatest impact on landslide susceptibility. This ensures the efficient utilization of limited resources for effective hazard management. Geospatial variable importance analysis contributes to informed infrastructure planning and design along the KKH. By understanding the key variables that influence landslide susceptibility, engineers and planners can incorporate appropriate measures to reduce the vulnerability of infrastructure to landslides. This includes considering geotechnical investigations, slope stabilization techniques, and route optimization to minimize exposure to high-risk areas. The findings from analyzing feature selection algorithms for geospatial variable importance can be applied beyond landslide susceptibility mapping. The knowledge gained can inform similar geospatial analysis tasks in other regions facing similar challenges. The insights obtained can guide researchers and practitioners in assessing the importance of variables for various geospatial phenomena and developing targeted mitigation strategies.

In summary, analyzing feature selection algorithms for geospatial variable importance in the context of landslide susceptibility mapping along the KKH offers several benefits. It enhances our understanding of landslide mechanisms, improves prediction and mapping accuracy, facilitates proactive hazard management, optimizes resource allocation, informs infrastructure planning, and provides valuable insights for broader geospatial analysis tasks. These benefits contribute to safer infrastructure, reduced risks, and more effective decision making for landslide-prone areas. Overall, by combining advanced feature selection algorithms with geospatial analysis, this study aims to provide a comprehensive understanding of the geospatial variable importance for landslide susceptibility mapping along the KKH, ultimately contributing to safer infrastructure and improved hazard management in the region. We also employ advance hyper optimization techniques to optimize the ANN model with the aim of maximizing the predictive capabilities of machine learning algorithms, using both hyperparameter and feature optimization techniques to give insight into complex phenomena, like landslides, and to optimize complex modeling techniques. It is essential to identify advance modeling techniques, such ensemble methods like Random Forests and Gradient Boosting, that are effective in handling the intricate interactions between different variables. They can capture nonlinear relationships, account for missing data, and highlight the significance of particular features. Geospatial data

analysis offers a valuable perspective. By integrating Geographic Information System (GIS) techniques, spatial relationships and topographical features can be incorporated into the analysis. Techniques like spatial statistics, including variogram analysis, can identify spatial autocorrelation, providing insight into patterns and correlations [14–16]. Advanced models can also be combined using ensemble methods for spatial analysis. Stacking or bagging, for example, allow for the amalgamation of distinct models, thereby leveraging the strengths of individual models to improve predictive accuracy. Deep Learning, specifically Convolutional Neural Networks (CNNs), comes into play when satellite or aerial imagery data are available. CNNs can extract pertinent features from images, capturing spatial patterns and texture information that are indicative of landslide susceptibility. Hybrid approaches that combine diverse data sources—such as remote sensing data, topographical data, and historical landslide records—can yield a comprehensive set of input features for the models being used in this study, enriching the understanding of susceptibility variables [17,18]. For situations where temporal dynamics are influential, models that incorporate both spatial and temporal features can provide a more accurate representation of landslide susceptibility changes over time. Moreover, Bayesian Network Models offer a probabilistic perspective, enabling the capture of intricate relationships among variables and providing a visualization of causal connections that contribute to landslide susceptibility. To strike a balance between predictive accuracy and interpretability, hybrid geostatistical models can merge geostatistical methods (e.g., kriging) with machine learning models, thereby considering spatial autocorrelation in predictions.

The main objective of this paper is to comprehensively investigate and improve the accuracy of landslide susceptibility mapping through the integration of various methodologies. This includes delineating the study area, identifying relevant landslide conditioning variables, generating susceptibility maps using the specified methodology, exploring different feature selection techniques to assess geospatial variables, and employing advanced hyperparameter techniques like Bayesian and Metaheuristic methods. The primary aim is to evaluate the effectiveness of these approaches and techniques in enhancing the accuracy of landslide susceptibility predictions, thereby contributing to a deeper understanding of landslide dynamics and potential mitigation strategies. This paper is structured into various sections. Section 2 delves into elucidating the study area and the variables contributing to landslides. In Section 3, the methodology employed for generating susceptibility maps is expounded upon. Section 4 elucidates the utilization of eight distinct feature selection techniques, each assessing geospatial variables based on criteria elucidated in detail within that section. Section 5 provides insight into the application of Bayesian and Metaheuristic hyperparameter techniques. The evaluation of the results obtained from our experiment is covered in Section 6.

### 1.1. Architecture of Artificial Neural Network

The architecture of the Artificial Neural Network (ANN) for landslide susceptibility mapping can be described as the input layer of the ANN, which receives the relevant data or features related to landslide susceptibility in the KKH region. These features may include variables such as geological characteristics, slope angles, aspect, geological properties, and land cover. The number of neurons specific to each setting, such as activation function, batch size, and epochs for each technique determined during the design and training of the ANN for BO_GP, BO_TPE, PSO, and GA, are mentioned in Table 1 below.

Table 1 represents the experimental results of different hyperparameter optimization techniques applied to ANN. Each row in the table corresponds to a different optimization technique, and the columns represent the evaluated performance metrics or hyperparameters.

The activation function is a crucial component of ANNs, as it introduces nonlinearity and affects the network's ability to model complex relationships. The table provides information about the activation functions used for each technique, such as "sgd", "tanh", and the value "1.742" for PSO.

**Table 1.** The information given includes the hyperparameters related to the optimization techniques (PSO, GA, BO_GP, and BO_TPE).

| HOP Technique | Activation | Batch Size | Epochs | Neurons |
|:---:|:---:|:---:|:---:|:---:|
| PSO | 1.742 | 0.367 | 40.273 | 95.430 |
| GA | sgd | 16 | 50 | 64 |
| BO_GP | tanh | 16 | 47 | 54 |
| BO_TPE | 1 | 16 | 50 | 80 |

Batch Size and Epochs: Batch size refers to the number of training samples used in each iteration of the optimization algorithm, while epochs represent the number of times the entire training dataset is passed through the network.

Neurons: Neurons refer to the number of units or nodes in each layer of the ANN. Table 1 shows the number of neurons used for each technique, such as 64 for GA.

The output layer of the ANN represents the predicted susceptibility to landslides in the KKH region. It generates a susceptibility map indicating the likelihood of landslides occurring at different locations within the study area. The backpropagation approach is commonly used to train ANNs. There are various training algorithms that can be applied, including gradient descent, conjugate gradient descent, robust, BFGS quasi-Newton, one-step secant, and Levenberg–Marquardt, and Bayesian regularization [19–21]. The choice of training algorithm may depend on the specific requirements and characteristics of the landslide susceptibility problem in the KKH region. The focus of this study is to evaluate the effectiveness of four optimization strategies in combination with ANNs. These strategies are PSO, GA, BO-GP, and BO-TPE. These optimization techniques are employed to fine-tune the hyperparameters of the ANN and enhance its performance in generating accurate landslide susceptibility maps. Overall, the architecture of the ANN involves an input layer that receives relevant data, hidden layers for processing and feature extraction, an output layer that produces the susceptibility map, training algorithms for optimizing the network's parameters, and optimization techniques to improve the performance of the ANN in generating accurate landslide susceptibility maps for the KKH region [22].

### 1.2. Benefits of Artificial Neural Networks

Artificial neural networks (ANNs) offer several benefits in various fields and applications. Here are some of the key benefits of using ANNs. Nonlinear mapping: ANNs excel at capturing and modeling complex, nonlinear relationships between input and output variables. They can learn and represent highly intricate patterns, making them effective in handling complex tasks that may not have clear linear relationships. Adaptive learning: ANNs have the ability to adapt and learn from examples or data. Through a process called training, ANNs can adjust their internal parameters to improve their performance on specific tasks. This adaptability allows ANNs to continuously learn and improve their predictions or classifications as new data becomes available. Parallel processing: ANNs can perform computations in parallel, making them suitable for handling large-scale and computationally intensive tasks [8,23,24]. This parallel processing capability enables ANNs to process multiple inputs simultaneously, leading to faster and more efficient computations. Fault tolerance and redundancy: ANNs have the ability to tolerate faults or errors in the system. Due to their distributed nature and interconnected structure, ANNs can still provide useful outputs even when some of the nodes or connections in the network are damaged or missing. This fault tolerance and redundancy makes ANNs robust in real-world scenarios where there may be noise or incomplete data. Pattern recognition and classification: ANNs are highly effective in pattern recognition and classification tasks [25–28]. They can learn and recognize complex patterns, allowing them to classify and categorize data into different classes or groups. This capability has numerous applications in fields such as image and speech recognition, natural language processing, and data

mining. Generalization: ANNs can generalize learned patterns to new, unseen data. Once trained on a representative dataset, ANNs can make accurate predictions or classifications on similar but previously unseen inputs. This generalization ability makes ANNs valuable in scenarios where new data needs to be processed or classified in real-time. Feature extraction: ANNs can automatically extract relevant features or representations from raw data, reducing the need for manual feature engineering. This capability is particularly useful when dealing with high-dimensional data, as ANNs can automatically learn and extract the most informative features to improve performance. Real-time processing: ANNs can be implemented in real-time systems, allowing for the quick and efficient processing of data. This capability is crucial in applications, such as real-time monitoring, control systems, and decision making, where immediate responses are required. These benefits demonstrate the versatility and power of artificial neural networks in solving complex problems like landslide susceptibility mapping and making accurate predictions or classifications across various domains.

## 2. Study Area

We conducted a study in the northern part of Pakistan, focusing on an approximately 332 km stretch of the KKH highway. The KKH highway is a major road that spans 1300 km, connecting various provinces of Pakistan, namely Punjab, Khyber Pakhtunkhwa, and Gilgit Baltistan, with Xinjiang, an autonomous region of China. Our study specifically covered the Gilgit, Hunza, and Nagar districts. Along the KKH, there are several villages including Juglot (located between 36°12′147″N latitude and 74°18′772″E longitude), Jutal, Rahimbad, Aliabad, and others, ultimately reaching Khunjarab Top, which serves as the border crossing between China and Pakistan. This region is situated alongside the Indus River, Hunza River, and Gilgit River.

The study area encompassed a length of 332 km and had a radius of 10 km, covering an area of 3320 km$^2$ along the KKH. The majority of the region consists of mountainous terrain, with its highest point reaching an elevation of 5370 m and the lowest point at 1210 m. Common natural hazards in this area include snow avalanches, landslides, and earthquakes Figure 1.

Technological advancements have propelled improvements in landslide modeling methods, encompassing the integration of remote sensing and GIS for comprehensive data analysis. Machine learning and AI algorithms aid in pattern recognition and predictive precision, while diverse datasets are fused to offer a holistic understanding of triggers and behavior [8,29]. Numerical simulation models enhance mechanics comprehension, and probabilistic approaches address uncertainties. Real-time analysis is enabled through big data and cloud computing, complemented by sensor networks for continuous monitoring. Crowdsourced data and early warning systems facilitate real-time data collection and alerts. The integration of climate change, decision support tools, and collaborative efforts elevate accuracy and preparedness, culminating in strides to mitigate landslide risks and impacts. A case study can showcase the application of advanced machine learning algorithms, including ANN, along with hyper optimization techniques, like Bayesian and metaheuristic methods, incorporating sophisticated feature selection methodologies for optimizing the modeling of complex phenomena, such as landslides [6].

The most common types of landslide in our study are debris flow, rockfalls and floods. Notable instances of devastating debris flows are the Attabad and Hunza landslides. Various rockfall events occurred on the Karakoram Highway, such as the rockfall incident that took place in Kohistan's Barseen area and Kohistan [30,31]. Streams can negatively impact the stability of slopes through two main mechanisms: undercutting due to toe erosion and saturation of the slide toe caused by increased water penetration. In the studied region, both the road and stream networks wield significant influence over the occurrence of landslides. Particularly, the area within a 100 m buffer zone between roads and watercourses demonstrates one of the highest Information Gain ratios, rivaled only by the slope (for further details, see Section 4). This trend is clarified by the fact that uncontrolled blasting

and excavation during road construction, as well as the establishment of stream channels for irrigation on these vulnerable slopes, frequently trigger land movements.

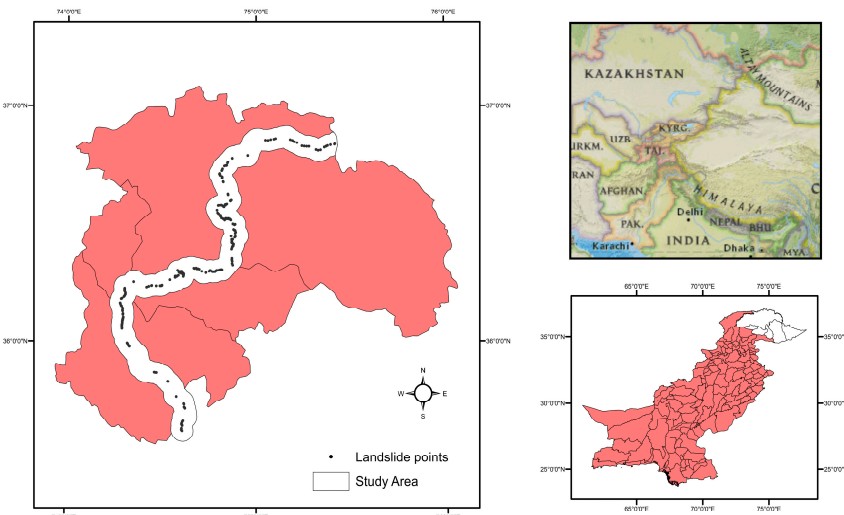

**Figure 1.** Our research focuses on the northern region of Pakistan. The KKH is place precisely where our study area is located.

As expected, areas covered by woodland, shrubland, and alpine pasture exhibit the lowest tendency of experiencing landslides. In a comparison between the landslide inventory and variables contributing to landslides, it becomes evident that roads, streams, and slope gradients hold the utmost importance as variables dictating the spatial distribution of landslides within the area. Gilgit receives an annual rainfall of approximately 154 mm. The region's irrigation depends on the flow of streams and rivers fed by the melting snow and glaciers from lofty mountains. The summer season is prolonged and brings higher temperatures. While the average wintertime temperature remains below 10 °C, occurrences of intense sunshine are infrequent, and temperatures rarely reach 40 °C (104 °F). The region's harsh weather conditions contribute to frequent instances of landslides and avalanches. These events are a result of the area's geologically unstable terrain and soils, which play a pivotal role in slope instability. Additionally, the steep mountain slopes make them susceptible to landslides as well.

*Geospatial Variables*

The geospatial variables used in our case study are as follows: slope, aspect, land cover, geology, precipitation, distance to faults, distance to streams, and distance to roads. The sources of the dataset are shown in Tables 2 and 3.

**Table 2.** Detail of data sources for geospatial variables used in our case study.

| Data | Variables | Scale/Resolution | Source |
|---|---|---|---|
| Sentinel 2 Satellite Images | Landslide inventory, LCLU, Road network | 10 m | |
| DEM | Slope Aspect Stream Network | 30 m | SRTM Shuttle Radar Topography Mission (USGS) United States Geological Survey |
| Geological Map | Geology Units and Fault lines | 30 m | Geological Survey of Pakistan |
| Google Earth Maps | Landslide Inventory Land Cover/Land Use Road Network | 2–5 m | |
| Field Survey | GPS Points | 1 m | |

**Table 3.** Geospatial variables used in our case study for landslide susceptibility mapping.

| Variables | Classes | Class Percentage % | Landslide Percentage % | Reclassification |
|---|---|---|---|---|
| Slope (°) | Very Gentle Slope < 5° | 17.36 | 21.11 | Geometrical interval reclassification |
| | Gentle Slope 5–15° | 20.87 | 28.37 | |
| | Moderately Steep Slope 15–30° | 26.64 | 37.89 | |
| | Steep Slope 30–45° | 24.40 | 10.90 | |
| | Escarpments > 45° | 10.71 | 1.73 | |
| Aspect | Flat (−1) | 22.86 | 7.04 | Remained unmodified (as is from source data). |
| | North (0–22) | 21.47 | 7.03 | |
| | Northeast (22–67) | 14.85 | 5.00 | |
| | East (67–112) | 8.00 | 11.86 | |
| | Southeast (112–157) | 5.22 | 14.3 | |
| | South (157–202) | 2.84 | 14.40 | |
| | Southwest (202–247) | 6.46 | 12.41 | |
| | West (247–292) | 7.19 | 16.03 | |
| | Northwest (292–337) | 11.07 | 11.96 | |
| Land Cover | Dense Conifer | 0.38 | 12.73 | |
| | Sparse Conifer | 0.25 | 12.80 | |
| | Broadleaved, Conifer | 1.52 | 10.86 | |
| | Grasses/Shrubs | 25.54 | 10.3 | |
| | Agriculture Land | 5.78 | 10.40 | |
| | Soil/Rocks | 56.55 | 14.51 | |
| | Snow/Glacier | 8.89 | 12.03 | |
| | Water | 1.06 | 16.96 | |
| Geology | Cretaceous Sandstone | 13.70 | 6.38 | |
| | Devonian–Carboniferous | 12.34 | 5.80 | |
| | Chalt Group | 1.43 | 8.43 | |
| | Hunza Plutonic Unit | 4.74 | 10.74 | |
| | Paragneisses | 11.38 | 11.34 | |
| | Yasin Group | 10.80 | 10.70 | |
| | Gilgit Complex | 5.80 | 9.58 | |
| | Trondhjemite | 15.65 | 9.32 | |
| | Permian Massive Limestone | 6.51 | 6.61 | |
| | Permanent Ice | 12.61 | 3.51 | |
| | Quaternary Alluvium | 0.32 | 8.65 | |
| | Triassic Massive Limestone and Dolomite | 1.58 | 7.80 | |
| | Snow | 3.08 | 2.00 | |
| Proximity to Streams (meter) | 0–100 | 19.37 | 18.52 | Geometrical interval reclassification |
| | 100–200 | 10.26 | 21.63 | |
| | 200–300 | 10.78 | 25.16 | |
| | 300–400 | 13.95 | 26.12 | |
| | 400–500 | 18.69 | 6.23 | |
| | >500 | 26.92 | 2.34 | |
| Proximity to Roads (meter) | 0–100 | 81.08 | 25.70 | |
| | 100–200 | 10.34 | 25.19 | |
| | 200–300 | 6.72 | 27.09 | |
| | 300–400 | 1.25 | 12.02 | |
| | 400–500 | 0.60 | 10.00 | |
| Proximity to Faults (meter) | 000–1000 | 29.76 | 27.30 | |
| | 2000–3000 | 36.25 | 37.40 | |
| | >3000 | 34.15 | 35.03 | |

Selecting and identifying the most influential geospatial variables is a challenging task, requiring significant ground reality understanding as well as understanding of the interdependence between the variables and variance and uncertainty that they hold therefor it's essential to carefully select the important variables and understand their importance



for predicting accurate landslide vulnerabilities in future [16,32,33]. Furthermore Table 3 and Figure 2 represent the data source of geospatial variable used in our case study.

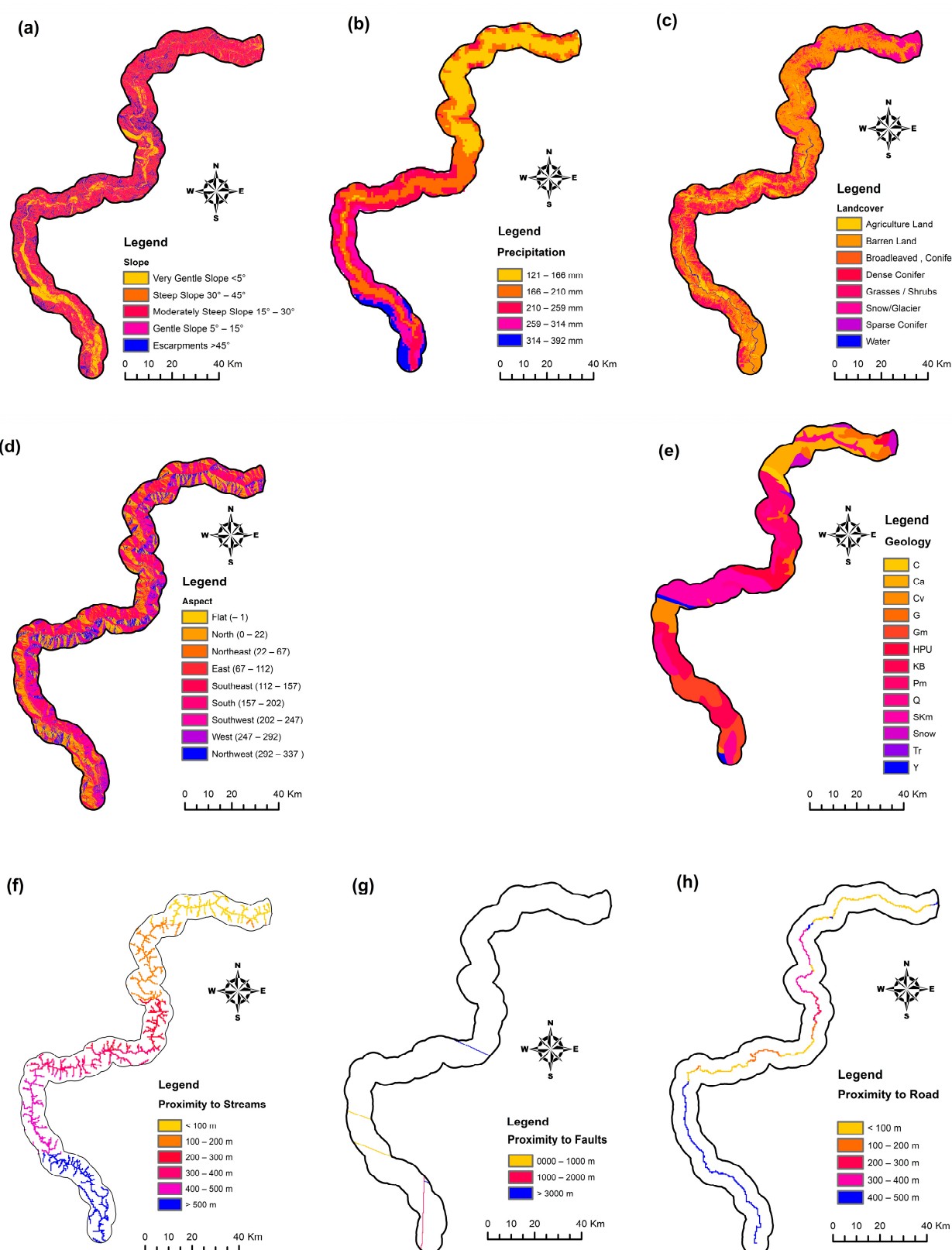

**Figure 2.** (**a**) Slope; (**b**) precipitation; (**c**) land cover; (**d**) aspect; (**e**) geology; (**f**) proximity to streams; (**g**) proximity to faults; (**h**) proximity to roads.

The categorization of influencing variables for landslide occurrences in our research area (Figure 2 and Table 3) involves four main categories: topological, hydrological, geological, and anthropological. Among these, slope and aspect are topological variables, with slope angle being the primary contributor to slope stability, while aspect-related variables, like sunlight exposure, winds, rainfall, soil moisture, and cracks, influence landslide frequency. Aspect and slope are determined using the SRTM DEM with a 30 m resolution and categorized into classes for analysis. Geological variables consider the geological map of Pakistan, including fault lines and various formations, as different geological units impact susceptibility to geomorphological processes. Hydrological variables encompass precipitation and proximity to waterways; this is significant due to rainfall-induced landslides in the study location. Anthropological elements include land usage and distance to highways, which are evaluated using Sentinel 2 images and supervised classification to analyze their effects on landslide movement. The map's accuracy was validated at 87%. Land use, road development, and construction activity are also considered as influencing variables on landslide stability. In total, eight variables—slope, aspect, land cover, geology, precipitation, distance to faults, distance to streams, and distance to roads—are examined in this case study (Figure 2).

## 3. Methodology

This study employed HOP techniques to search for the optimal hyperparameter configurations for the ANN models. This is important because selecting appropriate hyperparameters, such as activation function, batch size, epochs, and neuron configurations, can significantly impact the performance and generalization capabilities of ANNs. The four optimization techniques used in the study, namely PSO, GA, BO_GP, and BO_TPE, were applied to fine-tune the hyperparameters of the ANN models. Each technique employs a different approach to explore the hyperparameter space and find the best set of values. The performance of the ANN models was evaluated using AUC. Overall, the methodology involved applying different optimization techniques to search for the optimal hyperparameter configurations for the ANN models. The methodology aims to identify the most effective optimization approach and hyperparameter settings for accurately predicting the landslide susceptibility index. Additionally, our case study incorporated eight different techniques while considering various criteria, like multicollinearity, correlations, information sharing, variance, and uncertainty, to select landslide conditioning variables, as depicted in Figure 3.

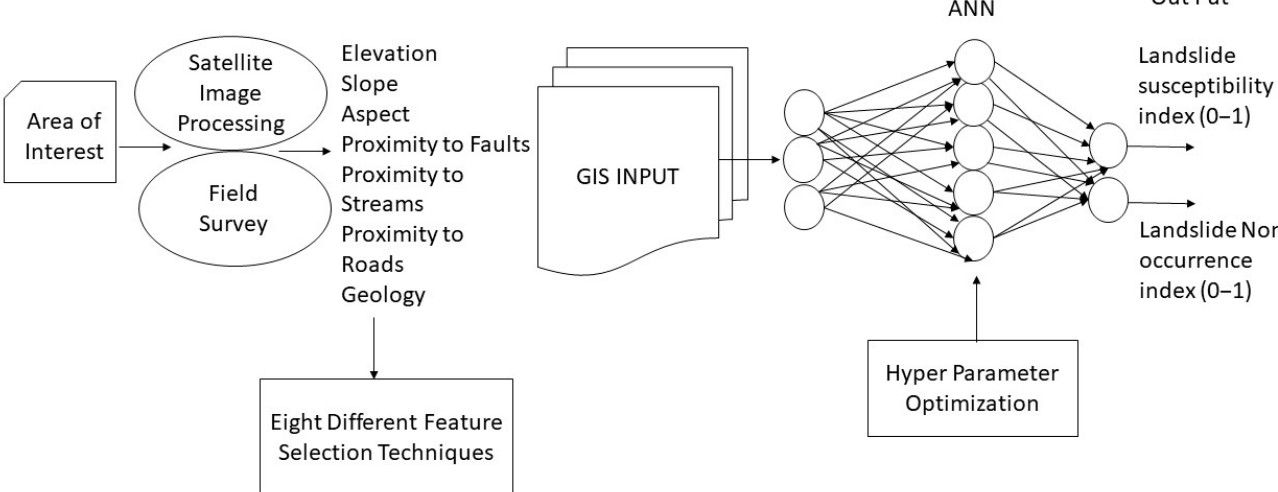

**Figure 3.** Methodology used for landslide susceptibility mapping along the KHH.

In complex tasks like predicting landslide susceptibility, where the relationships between variables can be intricate and data might be limited, it is essential to leverage all available tools to ensure model accuracy and reliability. Integrating feature selection and hyperparameter optimization allows us to build a more robust and effective model that can capture the complexities of the underlying processes. Combining feature selection and hyperparameter optimization in a methodology can provide several benefits for complex modeling tasks like predicting landslide susceptibility. For example, feature selection helps in identifying the most relevant features, which can prevent overfitting by reducing noise and irrelevant information. Hyperparameter optimization ensures that the model's complexity is controlled, further reducing the risk of overfitting. By selecting important features and optimizing hyperparameters, the model is more likely to generalize well to new, unseen data. This is crucial in predicting landslides, as the model needs to perform well on different terrains and conditions. Optimal hyperparameters and relevant features contribute to the model's performance. A well-tuned model with the right features can provide accurate predictions, which is vital for landslide susceptibility assessments. Combining these aspects can lead to better insights into the domain itself. Understanding which features are crucial and how hyperparameters affect model behavior can provide valuable information about the processes leading to landslides. Instead of exploring the entire hyperparameter space exhaustively, which can be very time-consuming, focusing on a reduced feature space due to feature selection can make the optimization process more efficient. Landslide susceptibility can vary across different regions and conditions. By optimizing both hyperparameters and feature selection, the model can be adapted and fine-tuned to different contexts. Including both feature selection and hyperparameter optimization can lead to more robust models. If certain features become less relevant due to changes in data distribution, the model's performance can still be maintained through proper hyperparameter tuning. Thus, the suggested methodology of combining feature selection and hyperparameter optimization in the context of predicting landslide susceptibility offers numerous benefits, particularly when dealing with complex modeling tasks, such as landslides. Overall, the combined approach brings synergy to the optimization process, leading to a more accurate, reliable, and efficient model for predicting landslide susceptibility.

## 4. Feature Selection Techniques

Feature selection is a vital process in machine learning and data analysis that involves choosing the most relevant features from a dataset while discarding irrelevant or redundant ones. It offers numerous benefits, including improved model performance by reducing overfitting and enhancing generalization; faster computation due to decreased data dimensionality; and increased interpretability, making models more understandable and usable. Additionally, it mitigates the challenges of high-dimensional data, simplifies model maintenance and updates, fosters better feature engineering practices, and supports efficient visualization. Feature selection also contributes to model robustness, cost savings by prioritizing valuable features, and alignment with algorithm assumptions, ultimately leading to more effective and practical applications of machine learning in various domains. Some of the most common and popular variable selection techniques are mentioned in our paper and are described in the subsequent subsections.

### 4.1. Information Gain

Information Gain (IG) is a measure used in decision tree algorithms and feature selection to assess the importance of a feature in a classification task. It quantifies the amount of information gained about the target variable by including a particular feature in the decision-making process.

Mathematically, Information Gain is calculated using the concept of entropy. Entropy measures the impurity or disorder of a set of instances in the dataset. The entropy E(C) and Information gain (IG) can be computed in relation to the conditioning variables of landslides and is defined as:

$$E(C) = -\sum_{i=1}^{j} P_i \log_2 P_i \tag{1}$$

In this context, $P_i$ denotes the ratio of the ith class within the complete dataset, and "m factor classes" encompass a set of values, i.e., $[v_1, v_2, \ldots, v_m]$.

$$IG(Y, X) = E(Y) - \sum_{v \in \{ F \mid 1,\ldots,m\}} \frac{|Y_v|}{Y} E(Y_v) \tag{2}$$

$$IG(Y, X) = E(Y) - E(Y \mid X) \tag{3}$$

The IG resulting from conditioning variables is represented as $IG(Y,X)$, while the anticipated weighted entropy is denoted as E(Y, X). Here, E(Y) represents the entropy of a landslide inventory Y containing j classes, while X, also with *n* classes, represents the conditioning variables of landslides. The value of E(Y) serves as an indicator of the uniformity within landslides in Y, and it is effectively utilized for selecting the most suitable conditioning factor.

The Information Gain value ranges from 0 to 1, with a higher value indicating that the feature provides more valuable information for the classification task. A higher Information Gain suggests that the feature helps in reducing the uncertainty or disorder in the dataset and contributes to better classification.

From Table 4 and Figure 4, "Roads" stand out as the most significant feature, with an Information Gain of 0.352. This indicates that it strongly aids in making accurate predictions. The "Faults" feature follows with an Information Gain of 0.195, making it a moderately influential predictor. "Streams" provide a reasonable level of information, contributing moderately with an Information Gain of 0.091. "Geology" and "Slope" possess modest Information Gain values of 0.039, implying that they offer some useful insights, though not as prominently as the top features. "Precipitation" has a lower Information Gain of 0.012, suggesting it contributes a limited amount of relevant information. "Aspect" and "Land Cover" exhibit no meaningful Information Gain (0.000), implying they have little to no impact on accurate predictions. In essence, the Information Gain values help prioritize the importance of each feature in facilitating accurate predictions or classifications, with "Roads" and "Faults" leading the way.

**Table 4.** Information Gain IG obtained for the geospatial variables.

| Feature | Information Gain |
|---|---|
| Roads | 0.352 |
| Faults | 0.195 |
| Streams | 0.091 |
| Geology | 0.039 |
| Slope | 0.039 |
| Precipitation | 0.012 |
| Aspect | 0.000 |
| Land Cover | 0.000 |

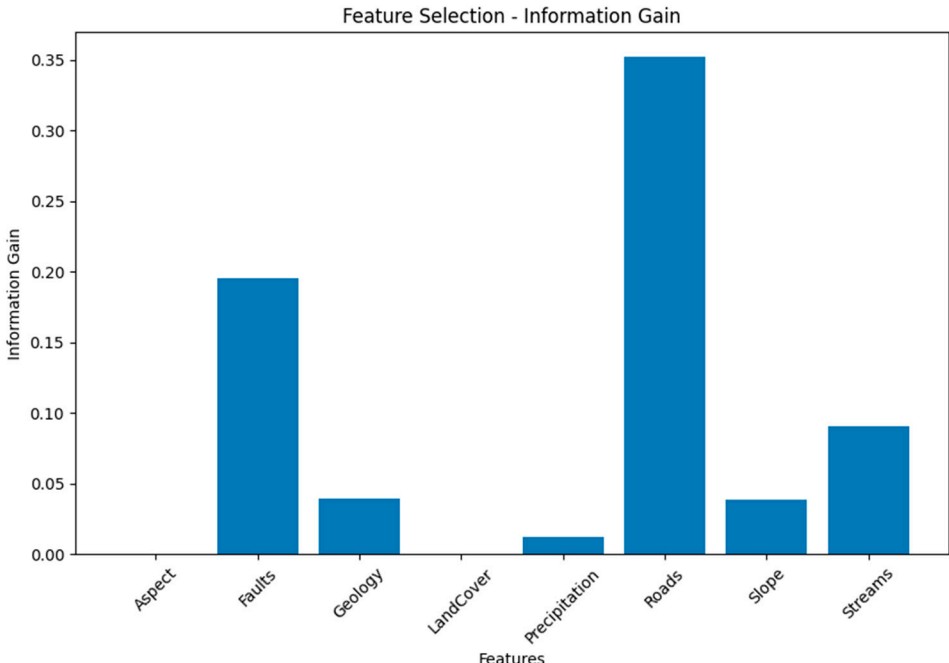

**Figure 4.** Information Gain IG obtained for the geospatial variables.

### 4.2. Variable Inflation Factor

The Variance Inflation Factor (VIF) is a measure used to assess the severity of multicollinearity among predictor variables in a regression model. It quantifies the extent to which the variance of the estimated regression coefficient for a specific predictor variable is inflated due to its correlation with other predictor variables.

Mathematically, the VIF for a predictor variable *i* is calculated as follows:

$$VIF_i = \frac{1}{1 - R_i^2} \tag{4}$$

where:

$VIF_i$ is the Variance Inflation Factor for predictor variable *i*;

$R_i^2$ is the coefficient of determination (R-squared) obtained by regressing variable *i* against all other predictor variables.

The VIF value ranges from 1 upwards, with values greater than 1 indicating the presence of multicollinearity. A VIF value of 1 indicates no multicollinearity, while values greater than 1 suggest increasing levels of multicollinearity. Generally, a VIF value of 5 or higher is often used as a threshold to identify problematic levels of multicollinearity, although the specific threshold may vary depending on the context and the field of study.

In practice, high VIF values indicate that the variance of the regression coefficient estimates for the predictor variable is inflated, which can lead to unstable and unreliable results. Multicollinearity can make it challenging to interpret the individual contributions and effects of predictor variables in the model. Remedial actions, such as excluding highly correlated variables or performing dimensionality reduction techniques, may be necessary to mitigate the issues caused by multicollinearity.

From Table 5 and Figure 5 "Land Cover" stands out with the highest VIF, indicating strong multicollinearity. Other features like "Precipitation", "Slope", "Geology", and "Aspect" also display varying degrees of multicollinearity. "Streams", "Faults", and "Roads" show relatively lower multicollinearity, making them potentially more independent in the analysis.

**Table 5.** Variable inflation factor VIF obtained for the geospatial variables.

| Feature | VIF |
| --- | --- |
| Land Cover | 18.090 |
| Precipitation | 7.167 |
| Slope | 6.276 |
| Geology | 4.986 |
| Aspect | 4.585 |
| Streams | 3.910 |
| Faults | 3.532 |
| Roads | 2.992 |

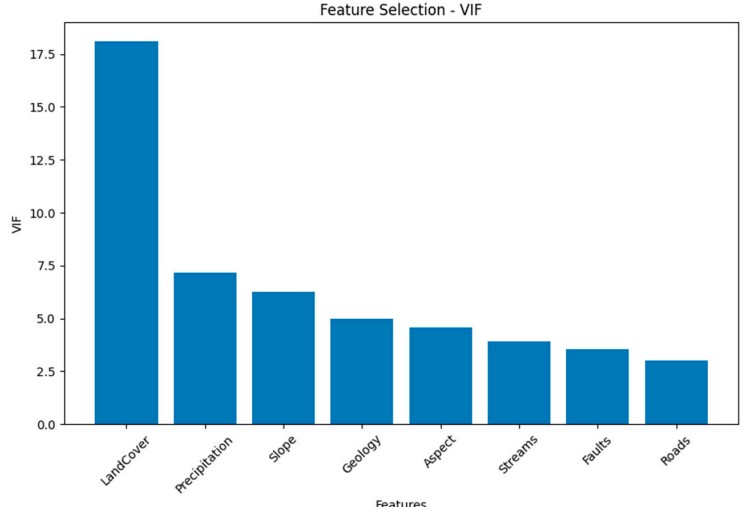

**Figure 5.** Variable inflation factor VIF obtained for the geospatial variables.

### 4.3. OneR Classifier

The OneR algorithm is a simple and interpretable classification algorithm. It generates a single rule for each predictor (attribute) in the data and selects the rule with the smallest total error. Each rule is based on the value of a single attribute and predicts the most frequent class for that attribute value. The OneR Classifier provides a straightforward way to assess the predictive power of individual attributes. The total error is calculated by summing the errors made by the rule for each attribute value. The formula for calculating the total error in OneR Classifier is dependent on the evaluation metric chosen, such as misclassification error rate, accuracy, or other relevant metrics.

**OneR Algorithm**
1. **for each predictor**,
2.    **for each value of that predictor, make a rule as follows**;
3.       Count how often each value of target (class) appears
4.       Find the most frequent class
5.       Make the rule assign that class to this value of the predictor
6.    Calculate the total error of the rules of each predictor
7. Choose the predictor with the smallest total error.

Assume that we have a dataset consisting of "n" instances, each represented by a feature "X" and a corresponding class label "Y". The One-R Classifier aims to find the best rule by selecting a single feature that minimizes the classification error. The rule is based on the mode (most frequent) class value for each unique value of the selected feature.

The following are the variables:

$X_i$ is the ith feature value;

$Y_i$ is the corresponding class label;
$f$ is the selected feature (attribute) for the rule;
C is the set of all possible class labels.

$$Rule: Y = argmaX\complement \in \complement P(Y = c | X = f(X_i)) \tag{5}$$

For each unique value of the selected feature $f(X_i)$, the classifier calculates the conditional probability of each class label C given that the feature has the value $f(X_i)$. It selects the class label with the highest conditional probability as the prediction for instances with that particular feature value.

Table 6 and Figure 6 summarize the accuracy scores for various features such as Roads, which is the most significant feature, achieving an accuracy score of 0.902 and therefore making it a strong predictor. "Faults" also stands out with an accuracy score of 0.707, indicating a notable predictive capability. "Streams" exhibit a decent accuracy score of 0.659, suggesting a meaningful predictive influence. "Land Cover" contributes with an accuracy score of 0.622, showcasing a moderate level of predictive power. These values emphasize the predictive strength of these four features, namely "Roads", "Faults", "Streams", and "Land Cover", in the context of the OneR Classifier.

**Table 6.** Feature accuracy score obtained for the geospatial variables using OneR Classifier.

| Feature Scores—OneR Classifier | |
|---|---|
| **Feature** | **Accuracy Score** |
| Roads | 0.902 |
| Faults | 0.707 |
| Streams | 0.659 |
| Land Cover | 0.622 |
| Aspect | 0.585 |
| Geology | 0.561 |
| Slope | 0.549 |
| Precipitation | 0.500 |
| Selected Feature: Roads Accuracy Score: 0.902 | |

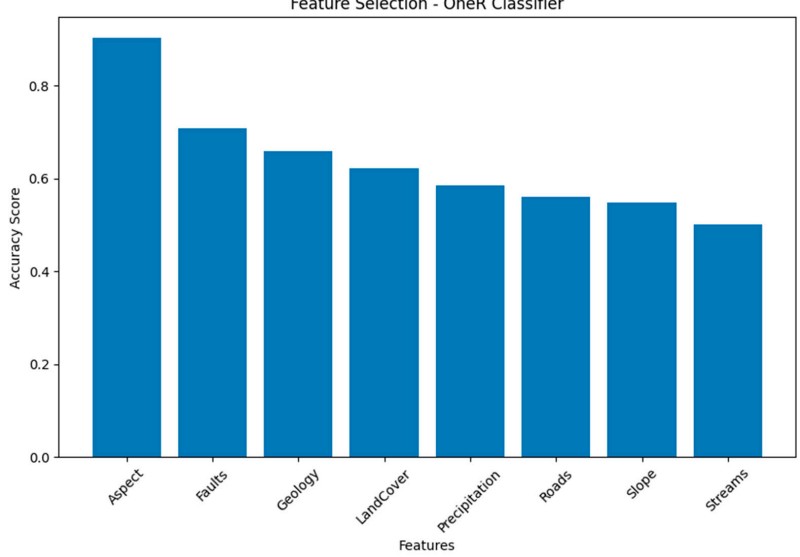

**Figure 6.** Feature accuracy score obtained for the geospatial variables using OneR Classifier.

### 4.4. Subset Evaluators

The principle behind the Correlation Feature Selection (CFS) measure is that it assesses feature subsets by considering the hypothesis that effective subsets should encompass features that are strongly correlated with the classification while maintaining minimal correlations among themselves. The subsequent equation provides the assessment of the quality of a feature subset S comprising k features:

$$MeritS_k = \frac{k\overline{r_{cf}}}{\sqrt{k + k(k-1)\overline{r_{ff}}}} \tag{6}$$

The symbol $\overline{r_{cf}}$ represents the average of all correlations between the features and the classification, while $\overline{r_{ff}}$ stands for the average of all correlations between the features themselves. The CFS criterion is formulated in the subsequent manner:

$$CFS = max_{S_k}\left[\frac{r_{cf_1} + r_{cf_2} + \cdots + r_{cf_k}}{k + 2\left(r_{f_1f_2} + r_{f_if_j} + \cdots + r_{f_kf_{k-1}}\right)}\right] \tag{7}$$

The variables $r_{f_if_j}$, $r_{cf_k}$ are termed as correlations; however, they do not specifically represent Pearson's correlation coefficient or Spearman's $\rho$. In Hall's dissertation [34], these correlations are not based on either of these methods. Instead, three distinct measures of relatedness are employed: minimum description length (MDL), Symmetrical Uncertainty, and relief. Correlation is explained in detail in Sections 4.5–4.7.

Assuming $x_i$ represents the indicator function for the membership of feature $f_i$ in a set, the previous statement can be reformulated into an optimization problem as follows:

$$CFS = max_{x\epsilon\{0,1\}^n}\left[\frac{\left(\sum_{i=1}^n a_i x_i\right)^2}{\sum_{i=1}^n x_i + \sum_{i\neq j} 2b_{ij}x_i x_j}\right] \tag{8}$$

Here, $x\epsilon\{0,1\}^n$ represents a binary vector x of length n, where each element $x_i$ is either 0 or 1. This vector encodes the selection of features in a subset, where 1 indicates that the feature is selected, while 0 indicates that it is not. $a_i$ is the coefficient associated with each feature. This coefficient quantifies the correlation between the ith feature and the classification. In other words, they represent how strongly the ith feature is related to the classification $b_{ij}$. This coefficient represents the correlations between pairs of features. The coefficient quantifies the correlations between different pairs of features, indicating how much they are correlated with each other. Subset Evaluators evaluate the performance of feature subsets by training a classifier on either the entire training dataset or a separate hold-out testing set. They assess the accuracy or other performance metrics of different subsets to determine the most informative features. Common Subset Evaluators include the evaluation of classification accuracy, F1 score, or other relevant metrics to measure the effectiveness of feature subsets. Subset Evaluators assess the performance of feature subsets by training a classifier on either the entire training dataset or a separate hold-out testing set.

From Table 7 and Figure 7 among the feature combinations tested, two stand out for their prominent accuracy scores: Geology and Roads. This combination yields the highest accuracy score of 0.927, showcasing strong predictive power. Similarly, the "Land Cover" and "Roads" combination also achieves an accuracy score of 0.927. The pairing of "Roads" and "Streams" maintains the same accuracy score of 0.927. Combining "Faults" and "Roads" results in a notable accuracy score of 0.915. The combination of "Aspect" and "Roads" follows with a substantial accuracy score of 0.902. These combinations exhibit the highest accuracy scores and thus stand out as the most prominent and predictive feature combinations in the context of the classification task.

**Table 7.** Feature combination accuracy score obtained for geospatial variables using subset evaluator.

| Feature Combination | Accuracy Score |
|---|---|
| Geology, Roads | 0.927 |
| Land Cover, Roads | 0.927 |
| Roads, Streams | 0.927 |
| Faults, Roads | 0.915 |
| Aspect, Roads | 0.902 |
| Roads, Slope | 0.878 |
| Precipitation, Roads | 0.854 |
| Faults, Streams | 0.793 |
| Faults, Precipitation | 0.780 |
| Faults, Geology | 0.744 |
| Faults, Land Cover | 0.744 |
| Faults, Slope | 0.732 |
| Aspect, Faults | 0.720 |
| Geology, Land Cover | 0.720 |
| Geology, Precipitation | 0.720 |
| Geology, Streams | 0.707 |
| Aspect, Streams | 0.683 |
| Aspect, Precipitation | 0.671 |
| Aspect, Slope | 0.671 |
| Land Cover, Streams | 0.634 |
| Precipitation, Streams | 0.634 |
| Aspect, Geology | 0.622 |
| Land Cover, Precipitation | 0.622 |
| Slope, Streams | 0.622 |
| Geology, Slope | 0.610 |
| Land Cover, Slope | 0.610 |
| Aspect, Land Cover | 0.598 |
| Precipitation, Slope | 0.573 |

*4.5. Relief Attribute Evaluator*

The Relief algorithm evaluates the worth of an attribute by sampling instances and comparing the attribute values of the nearest instances from the same and different classes. It measures the attribute's ability to distinguish between classes based on the differences in attribute values for nearby instances. The Relief Attribute Evaluator is commonly used in feature selection for classification tasks, particularly for handling imbalanced datasets. The Relief algorithm evaluates the worth of an attribute by sampling instances and comparing the attribute values of the nearest instances from the same and different classes. Relief computes the weight for each attribute based on the differences in attribute values between the nearest instances and Is an algorithm for feature weighting that demonstrates sensitivity to interactions among features. It strives to estimate the ensuing difference in probabilities related to the weight of a feature X, where

$$W_X = P(\text{different value of X}|\text{ nearest instance of different class})$$
$$- P(\text{different value of X}|\text{ nearest instance of same class})$$

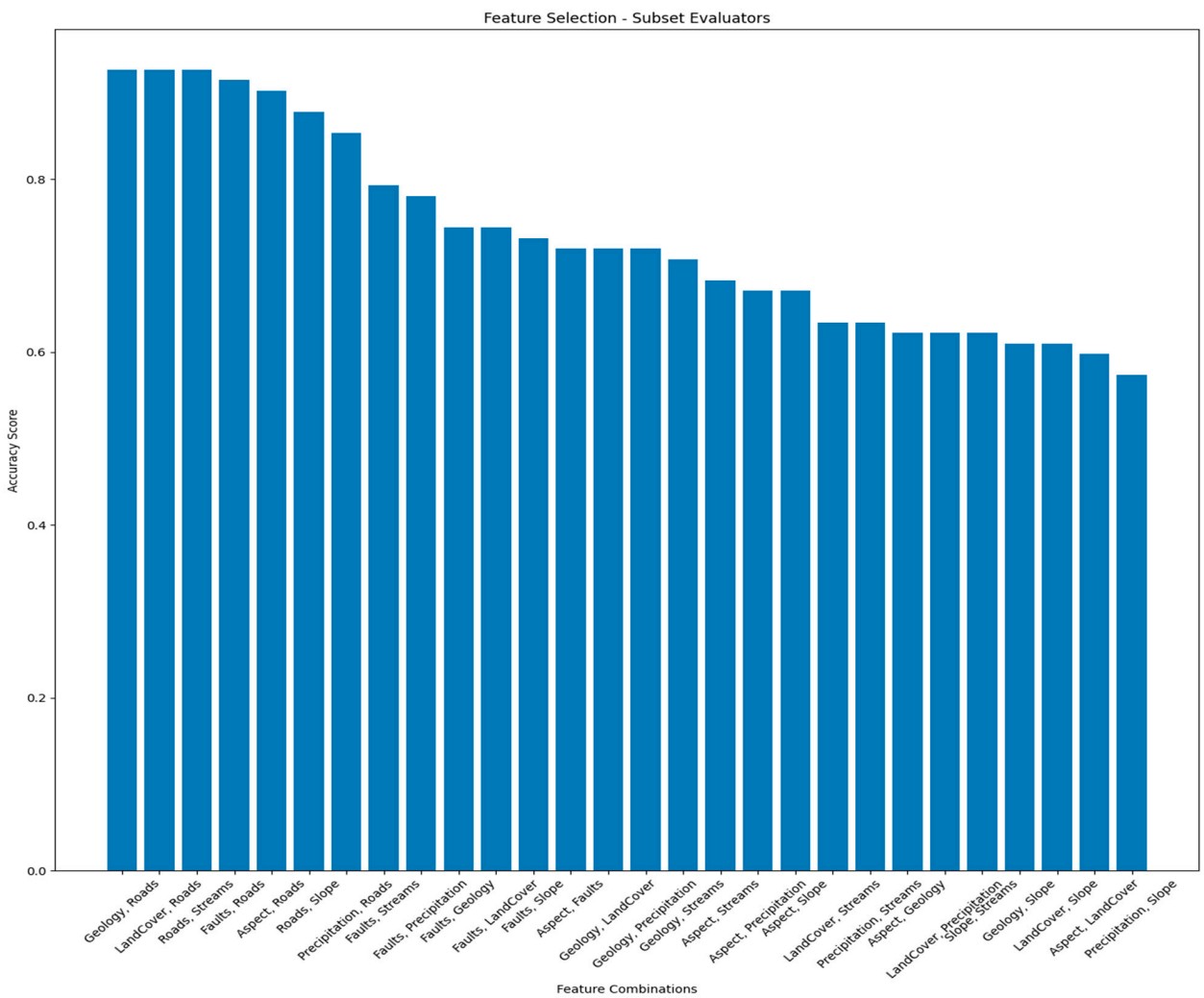

**Figure 7.** Feature combination accuracy score obtained for geospatial variables using subset evaluator.

By eliminating the contextual sensitivity introduced by the "nearest instance" condition, attributes are treated as mutually independent.

$$Relief_X = P(\text{different value of X}|\text{ different class})$$
$$-P(\text{different value of X}|\text{ same class})$$

which can be reformulated as

$$Relief_X = \frac{\text{Ginì } \times \sum_{x \in X} p(x)^2}{(1 - \sum_{c \in C} p(c)^2) \sum_{c \in C} p(c)^2} \tag{9}$$

where C is the class variable, and $\sum_{c \in C} p(c)^2$ calculates the sum of probabilities squared for each class. It reflects the concentration of instances in each class, and $\sum_{x \in X} p(x)^2$ calculates the sum of probabilities squared for each possible value of attribute X. It reflects how concentrated the distribution of attribute values is.

$$\text{Ginì} = \left[ \sum_{c \in C} p(c)(1 - p(c)) \right] - \sum_{x \in X} \frac{p(x)^2}{\sum_{x \in X} p(x)^2} \sum_{c \in C} p(c|x)(1 - p(c|x)) \tag{10}$$

Ginì represents a modification of an alternative attribute quality measure known as the Gini-index2. Both Ginì and the Gini-index share similarities with Information Gain, as they both exhibit a bias towards attributes with a greater number of values. For the

symmetrical utilization of the relief method with two features, the measurement can be computed twice (with each feature considered as the "class" in turn), and then the results can be averaged.

From Table 8 and Figure 8, the Relief Attribute Evaluator highlights the variable importance of different geospatial features, with "Roads", "Geology", and "Faults" being the most prominent contributors with an importance of 0.227, 0.156, 0.118 to the analysis, respectively.

**Table 8.** Geospatial variable importance obtained from Relief Attribute Evaluator.

| Feature | Importance |
|---|---|
| Roads | 0.227 |
| Geology | 0.156 |
| Faults | 0.118 |
| Streams | 0.112 |
| Precipitation | 0.060 |
| Aspect | 0.050 |
| Slope | 0.035 |
| Land Cover | 0.015 |

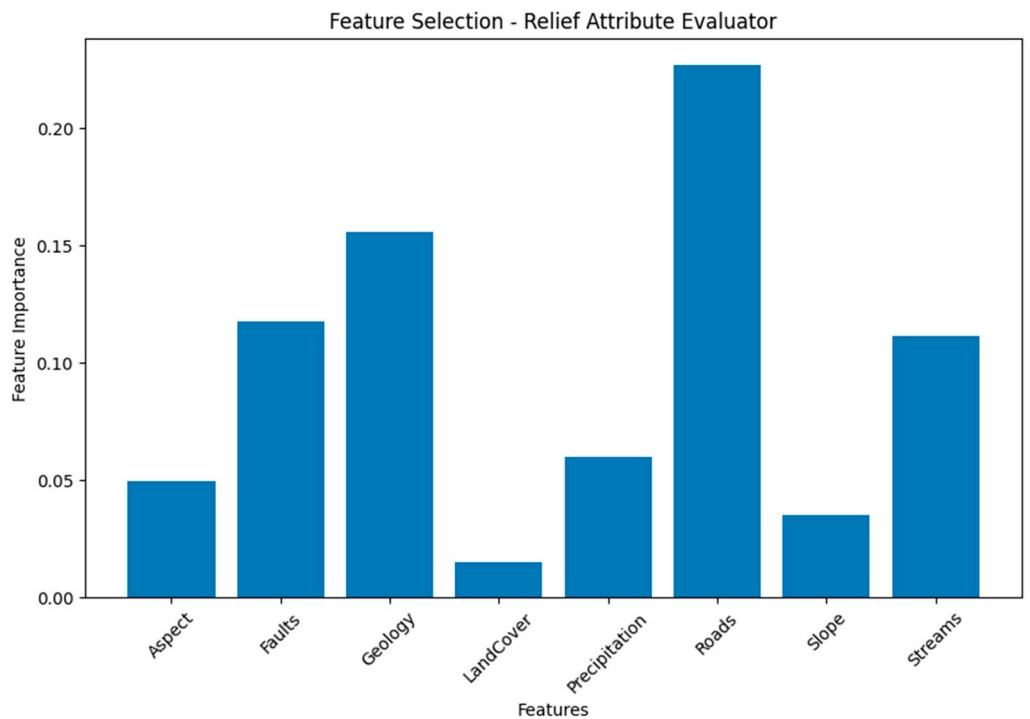

**Figure 8.** Geospatial variable importance obtained from Relief Attribute Evaluator.

*4.6. Correlation*

Correlation measures the linear relationship between an attribute and the class variable. Pearson's correlation coefficient is a common measure used to assess the strength and direction of the linear association. Higher absolute correlation values indicate a stronger relationship between the attribute and the class variable. Correlation analysis helps identify attributes that exhibit a significant relationship with the target variable.

Correlation measures the linear relationship between an attribute and the class variable. Pearson's correlation coefficient is commonly used to assess the strength and direction of the linear association. The formula for Pearson's correlation coefficient is:

Correlation = Cov(x, y)/(StdDev(x) * StdDev(y))

where

Cov(x, y) is the covariance between the attribute x and the class variable y;
StdDev(x) is the standard deviation of attribute x;
StdDev(y) is the standard deviation of the class variable y;
Formula for Pearson Correlation Coefficient:

$$\gamma_{xy} = \frac{\sum_{i=1}^{n}(x_{i-\bar{x}})(y_i - \bar{y})}{\sqrt{\sum_{i=1}^{n}(x_{i-\bar{x}})}\sqrt{\sum_{i=1}^{n}(y_i - \bar{y})^2}} \tag{11}$$

Here, n represents the sample size, while $x_i$ and $y_i$ denote individual sample points indexed by i, $\bar{x} = \frac{1}{n}\sum_{i=1}^{n}(x_i)$ (the sample mean), similarly for $\bar{y}$.

Rearranging leads us to the following formula for the correlation coefficient $\gamma_{xy}$:

$$\gamma_{xy} = \frac{n\sum x_i y_i - \sum x_i \sum y_i}{\sqrt{n\sum x_i^2 - (\sum x_i)^2}\sqrt{n\sum y_i^2 - (\sum y_i)^2}} \tag{12}$$

Rearranging again give us:

$$\gamma_{xy} = \frac{\sum_i x_i y_i - n\overline{xy}}{\sqrt{\sum x_i^2 - n\bar{x}^2}\sqrt{\sum y_i^2 - n\bar{y}^2}} \tag{13}$$

An alternative representation provides the formula for $\gamma_{xy}$ as the average of the products of the standardized scores, as shown below:

$$\gamma_{xy} = \frac{1}{n-1}\sum_{i=1}^{n}\left(\frac{x_i - \bar{x}}{s_x}\right)\left(\frac{y_i - \bar{y}}{s_y}\right) \tag{14}$$

where n, $x_i$, $y_i$, $\bar{x}$, and $\bar{y}$ are defined above, while $s_x$ and $s_y$ are defined as $s_x = \sqrt{\frac{1}{n-1}\sum_{i=1}^{n}(x_i - \bar{x})^2}$, the sample standard deviation, similarly for $s_y$.

Table 9 and Figure 9 provides a ranking of geospatial variables based on their correlation with the target variable, with Aspect being the most important and Roads being the least important regarding correlation with the class variable this information can be used to prioritize variables for further investigation or modeling in their geospatial analysis.

**Table 9.** Geospatial variable importance obtained from Correlation.

| Feature | Importance |
|---|---|
| Aspect | 1.000 |
| Geology | 0.100 |
| Precipitation | 0.090 |
| Faults | 0.066 |
| Land Cover | 0.043 |
| Streams | 0.022 |
| Slope | 0.020 |
| Roads | 0.005 |

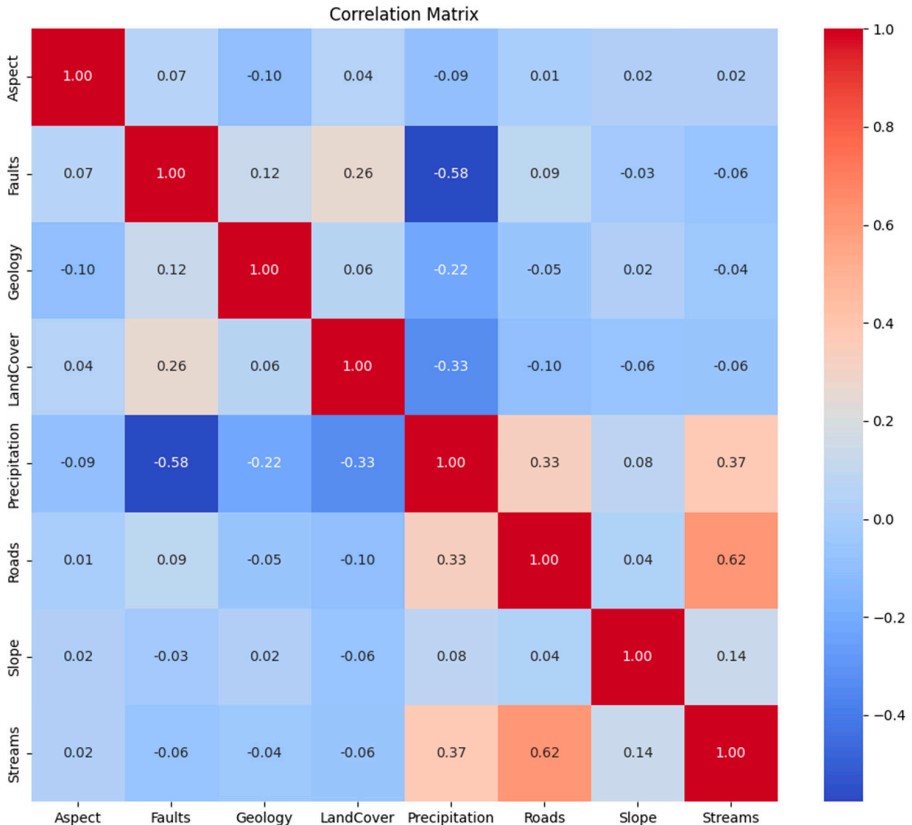

**Figure 9.** Geospatial variable importance obtained from Correlation.

### 4.7. Symmetrical Uncertainty

Symmetrical Uncertainty is an information-theoretic correlation measure based on entropy. It quantifies the amount of shared information between an attribute and the class variable. The measure takes into account both the attribute's predictability of the class and the class's predictability of the attribute. Symmetrical Uncertainty allows for the assessment of the mutual dependence between attributes and the target variable, enabling the identification of informative attributes.

The formula for Symmetrical Uncertainty is:

Symmetrical Uncertainty = (2 * Mutual Information)/(Entropy(X) + Entropy(Y))

where

Mutual Information measures the amount of shared information between the attribute X and the class variable Y;

Entropy(X) is the entropy of attribute X;

Entropy(Y) is the entropy of the class variable Y;

Normalized forms of mutual information can be derived using coefficients of constraint [35], uncertainty coefficients [36,37], or proficiency measures:

$$C_{XY} = \frac{I(X;Y)}{H(Y)} \text{ and } C_{YX} = \frac{I(X;Y)}{H(X)}$$

The values of the two coefficients lie within the range of 0 to 1, but they might not be identical. In certain situations, there could be a preference for a symmetric measurement, like the subsequent redundancy measure.

$$R_{max} = \frac{\min\{H(X), H(X)\}}{H(X) + H(Y)} \tag{15}$$

When one variable becomes entirely superfluous given the information about the other variable. Refer also to redundancy in information theory. An additional measure that exhibits symmetry is the symmetric uncertainty [38], which is defined as follows:

$$U(X,Y) = 2R = 2\,\frac{I(X;Y)}{H(X) + H(Y)} \tag{16}$$

This corresponds to the harmonic mean of the two uncertainty coefficients, denoted as $C_{XY}$ and $C_{YX}$. If we view mutual information as a specific instance of total correlation or dual total correlation, their respective normalized versions are as follows:

$$\frac{I(X;Y)}{\min[H(X), H(X)]} \; and \; \frac{I(X;Y)}{H(X,Y)} \tag{17}$$

This standardized variant is also recognized as the Information Quality Ratio (IQR) [39], which assesses the quantity of information in one variable concerning another variable in relation to the overall uncertainty.

$$IQR(X,Y) = E[I(X;Y)] = \frac{I(X;Y)}{H(X,Y)} = \frac{\sum_{x \epsilon X} \sum_{y \epsilon Y} p(x,y) \log p(x)p(y)}{\sum_{x \epsilon X} \sum_{y \epsilon Y} p(x,y) \log p(x,y)} \tag{18}$$

A normalization process [40,41] can be derived by initially considering mutual information as an analogy to covariance (where Shannon entropy is comparable to variance). Subsequently, the normalized mutual information is computed in a manner similar to the Pearson correlation coefficient.

$$\frac{I(X;Y)}{\sqrt{H(X)H(Y)}}$$

Each of these feature selection techniques offers a unique approach to assess the importance and relevance of geospatial variables for landslide susceptibility mapping. They provide different perspectives and criteria to evaluate the impact of variables on the target variable, allowing researchers and practitioners to gain valuable insights into the most influential variables affecting landslide occurrences along the Karakoram Highway.

From Table 10 and Figure 10 we can observed that the Symmetrical Uncertainty values provide insights into the degree of association between each geospatial variables. "Roads" and "Faults" appear as the most associated features, followed by "Streams" and "Geology", while "Precipitation" and "Aspect" show minimal to no association.

**Table 10.** Symmetrical Uncertainty obtained for the geospatial variables.

| Feature | Symmetrical Uncertainty |
|:---:|:---:|
| Roads | 0.683 |
| Faults | 0.435 |
| Streams | 0.247 |
| Geology | 0.128 |
| Land Cover | 0.093 |
| Slope | 0.053 |
| Precipitation | 0.001 |
| Aspect | 0.000 |

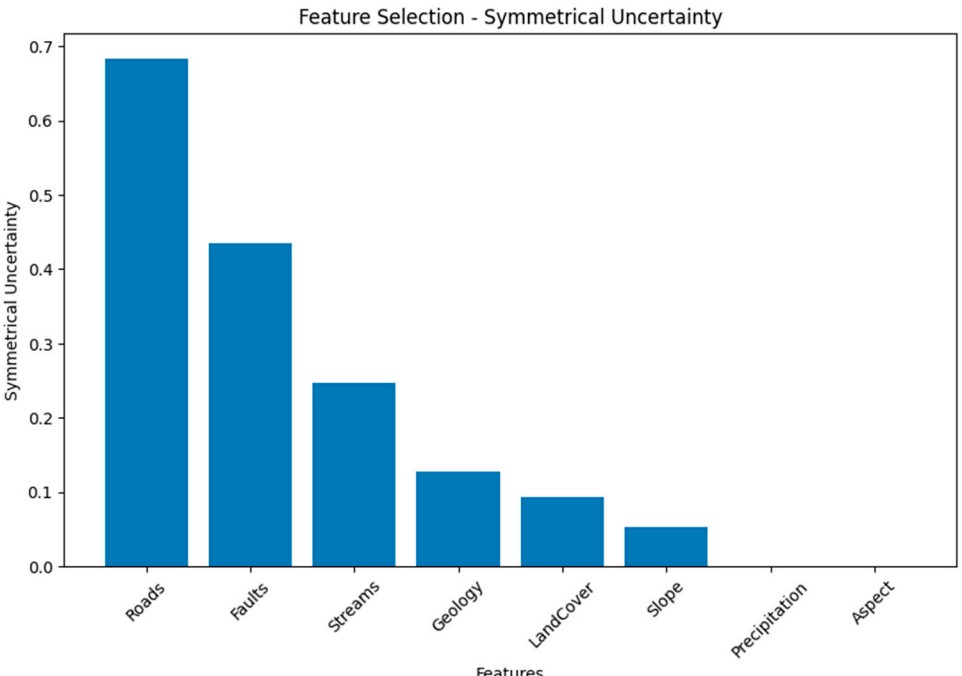

**Figure 10.** Symmetrical Uncertainty obtained for the geospatial variables.

*4.8. Principal Components*

Principal Components Analysis (PCA) is a dimensionality reduction technique that transforms the original features into a new set of uncorrelated variables called principal components. These components are ordered based on the amount of variance they explain in the data. PCA helps identify the most important features by considering those associated with the highest-variance principal components, thereby reducing the dimensionality of the dataset.

Mathematically, the transformation is defined by a set of size $\iota$ of p-dimensional weight vectors or coefficients $w_{(k)} = \left( w_1, \ w_2 \ldots . w_p \right)_{(k)}$ that correspond to each row vector $x_i$ of matrix X. These weight vectors map each $x_i$ to a new vector of principal component scores following the form $t_i = (t_i, \ldots t_\iota)_i$, as shown below.

$$t_{k_{(i)}} = X_i \cdot W_{(k)} \tag{19}$$

To optimize variance maximization, the initial weight vector $W_{(1)}$ must adhere to the condition:

$$W_{(1)} = \arg \max_{||W||=1} \left\{ \sum_i t_{1(i)}^2 \right\} = \arg \max_{||W||=1} \left\{ \sum_i (X_i . w)^2 \right\} \tag{20}$$

Alternatively, expressing this in matrix notation yields

$$W_{(1)} = \arg \max_{||W||=1} \left\{ ||Xw||^2 \right\} = \arg \max_{||W||=1} \left\{ \frac{w^T X^T X w}{w^T w} \right\} \tag{21}$$

The quantity aimed for maximization can be identified as a Rayleigh quotient. A well-known outcome for a positive semidefinite matrix like $X^T X$ is having the upper limit of the quotient be the matrix's largest eigenvalue, which si achieved when w is the corresponding eigenvector. Once $W_{(1)}$ is determined, the initial principal component of a data vector can be expressed either as a score, $t_{1_i} = X_{(i)} . W_{(1)}$, in the transformed coordinates, or as the corresponding vector in the original variables, $\{X_{(i)} . W_{(1)}\} W_{(1)}$.

The k-th component can be computed by subtracting the sum of the first $k-1$ principal components from the matrix X, such that

$$\hat{X}_k = X - \sum_{s=1}^{k-1} X w_s \, w_{(s)}^T \tag{22}$$

and then determining the weight vector that captures the highest variance from this modified data matrix, where

$$W_{(k)} = \arg\max_{||W||=1} \left\{ \left|\left| \hat{X}_k w \right|\right|^2 \right\} = \arg\max \left\{ \frac{w^T \hat{X}_k^{\,T} \hat{X}_k w}{w^T w} \right\} \tag{23}$$

Hence, the k-th principal component of a data vector $X_{(i)}$ can be expressed as a score, where $t_{1_i} = X_{(i)} \cdot W_{(1)}$ in the transformed coordinates. Alternatively, it can be represented as the corresponding vector in the original variable space, $\left\{ X_i \cdot W_{(k)} \right\} W_{(k)}$, where $W_{(k)}$ signifies the kth eigenvector of the matrix $X^T X$.

Consequently, the complete decomposition of X into principal components can be represented as:

$$T = XW \tag{24}$$

Here, W is a p-by-p matrix of weights, with its columns representing the eigenvectors of $X^T X$. The transpose of W is occasionally referred to as the whitening or sphering transformation. The columns of W, scaled by the square root of corresponding eigenvalues—effectively eigenvectors amplified by variances—are known as loadings in PCA or factor analysis.

$X^T X$ can be identified as being proportional to the empirical sample covariance matrix of the dataset $X^T$. The sample covariance "Q" between two distinct principal components across the dataset is defined as

$$\begin{aligned} Q\!\left( PC_{(j)}, PC_{(k)} \right) &\alpha \left( X w_j \right)^T (X w_k) \\ &= w_j^T X^T X w_k \\ &= w_j^T \lambda_k w_k \\ &= \lambda_k w_j^T w_k \end{aligned} \tag{25}$$

Expressed in matrix notation, the empirical covariance matrix for the original variables can be formulated as

$$Q\alpha \, X^T X = w \Lambda w^T \tag{26}$$

The empirical covariance matrix among the principal components takes on the following form:

$$w^T Q w \alpha w^T w \Lambda w^T w = \Lambda$$

Here, $\Lambda$ represents the diagonal matrix comprising the eigenvalues $\lambda_k$ of $X^T X$. Each $\lambda_k$ is the sum of squared values across the dataset for component k; specifically, $\lambda_k = \sum_i t_{k(i)}^2 = \sum_i (X_i \cdot w_k)^2$.

Mapping T = XW converts a data vector $X_i$ from an initial p-variable space to a fresh p-variable space, in which the variables are uncorrelated across the dataset. However, it is not obligatory to retain all the principal components. Selecting solely the first L principal components, derived by employing only the initial L eigenvectors, results in the truncated transformation:

$$T_L = X w_L \tag{27}$$

Here, the matrix $T_L$ now possesses n rows while containing only L columns. In simpler terms, PCA grasps a linear transformation, $t = w_L^T x$, $x \epsilon R^p$, $t \epsilon R^L$, where x belongs to the p-dimensional space of real numbers, and t belongs to the L-dimensional space

of real numbers. In this equation, the columns of the p × L matrix $w_L$ constitute an orthogonal basis for the L features (the elements of the representation t), ensuring their lack of correlation. Through its design, among all the transformed data matrices containing only L columns, this score matrix maximizes the retained variance from the initial data while simultaneously minimizing the aggregate squared reconstruction error.

$$||Tw^T - T_L w_L^T||_2^2 \text{ or } ||X - X_L||_2^2$$

Dimensionality reduction like this is an immensely beneficial step for visualizing and handling datasets with high dimensions, all while retaining the maximum possible variance within the dataset. For instance, when L = 2 is chosen and solely the first two principal components are retained, it results in a two-dimensional plane amidst the high-dimensional dataset in which the data's dispersion is maximized. Consequently, if the data contains clusters, they will also maximally spread out, making them more distinguishable when plotted on a two-dimensional graph. In contrast, if two directions through the data or two original variables are chosen arbitrarily, the clusters might be far less separated and might even substantially overlap, rendering them practically indistinguishable.

Table 11 and Figure 11 shows variance ratios that tell how much of the total variability in your dataset is explained by each principal component. The first principal component (PC1), which corresponds to Aspect, captures the largest portion of variation, indicating that Aspect is the most dominant variable in terms of explaining the data's variability. The subsequent components represent decreasing amounts of variance, with Streams having the least impact on the overall variation.

**Table 11.** Principal Component Analysis (PCA) for geospatial variables used in our experiment.

| Components | Variance Ratio |
|---|---|
| Aspect | 27.22% |
| Fault | 18.17% |
| Geology | 13.66% |
| Land Cover | 12.50% |
| Precipitation | 10.57% |
| Roads | 9.70% |
| Slope | 4.71% |
| Streams | 3.48% |

In conclusion, we can observe a consistent presence and importance of the "Roads" feature in all feature selection techniques mentioned above. "Roads" demonstrates several favorable characteristics across different techniques, including high Information Gain; low multicollinearity, as indicated by a low VIF; high accuracy scores in the OneR Classifier; inclusion in high-accuracy feature combinations; and relatively high importance scores from the Relief Attribute Evaluator and for Symmetrical Uncertainty. These findings suggest that "Roads" holds potential significance and influence in the analysis, regardless of the specific evaluation technique or metric employed. Additionally, the analysis provides further insights into other features. For example, "Fault" and "Streams" show relatively high Information Gain, "Aspect" has a high variance ratio in the PCA components, and "Aspect" and "Slope" display lower importance scores and correlation with the target variable. However, to make a final determination, it is recommended that a comprehensive analysis be conducted, including assessing multicollinearity; evaluating the contributions of other features; and considering additional factors, such as interpretability and practicality. Such analysis will help determine the optimal set of variables for accurate and reliable landslide susceptibility mapping.

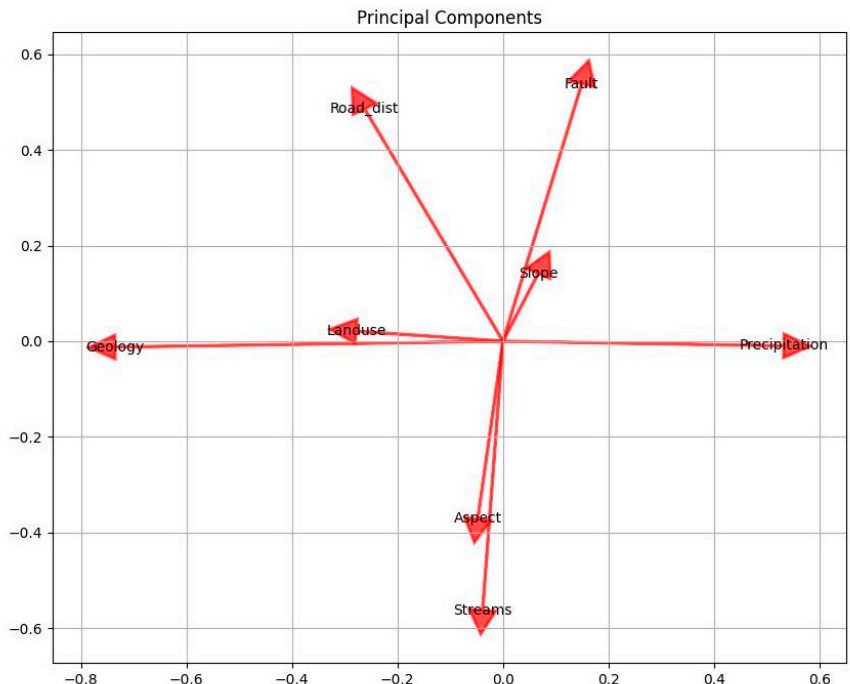

**Figure 11.** The covariance matrix is computed to understand the relationships between the features in the dataset. It represents how each feature changes with respect to the others.

It can also be evident from ground surveys that the roads are constructed in harsh mountainous terrain with improper blasting that makes the slope very vulnerable near the roads, where most of the landslides occur. The second most influential factor that we identified from our analysis is "Fault". This factor appears in multiple feature selection techniques (Section 4) with notable characteristics. It has relatively high Information Gain, moderate importance scores, and is included in high-accuracy feature combinations in the subset evaluator table. While it may not be as consistently influential as "Roads", it still demonstrates significance in the analysis. And this fact can also be evident from ground survey, which highlighted the fact that most landslide happened around the region near fault lines. The "Streams" factor also appears in multiple tables, showing relatively high Information Gain and being included in high-accuracy feature combinations. It demonstrates importance in the analysis, although to a lesser extent compared to "Roads" and "Fault". Again this phenomena can be verified from our case study, in which most of the land is irrigated from the streams the emerge from rivers that are filled with water from the melting snow and glaciers from high mountains, and even the man-made stream channel makes the region more venerable to landslides. Other features, while not as consistently influential as the three variables mentioned above, show importance in specific tables. For example, "Geology" and "Aspect" have relatively high importance scores in the Relief Attribute Evaluator table, and "Land Cover" has notable importance in the subset evaluator table. It is important to note that the level of influence may vary depending on the specific analysis techniques and evaluation measures used eight different techniques used in our paper. Additionally, the interpretation of influence should consider the context and objectives of the analysis. Further analysis, considering the dataset and specific requirements, is recommended to finalize the most influential variables for the specific task at hand. As we understand, diminishing entropy can lead to a loss of information, potentially causing important patterns to be forfeited [42]. Therefore, it is essential to carefully consider every variable.

## 5. Metaheuristic and Bayesian Algorithms

### 5.1. Genetic Algorithms

Although it is difficult to guarantee that the ANN can always generalize the testing data successfully, using a genetic method to avoid the model from getting stuck in a local minimum situation could assist in increasing accuracy rates. The three main components of the Genetic Algorithm (GA) are crossover, mutation, and selection. To realize elitism, the process first chooses the elite parents for the gene pool (a list that monitors the best weighting matrix). The crossover is then put into practice. The method chooses two genes at random out of the best genes (weighted matrix) and recombines them according to a specific strategy defined in the accompanying Python code. For instance, we chose a split point for the elite genes 1 and 2 at random in this instance. We then combine the second portion of gene 2 with the first part of gene 1, and for the remaining pieces of the two genes, we do the reverse procedure. We have two possibly elite genes that were recombined as a result. Then, given that it occurs at random, a mutation could happen. After completing the crossover for each generation, the mechanism will produce a random integer between 0 and 1. A specific area of the weighted matrix, which is likewise created randomly, will be multiplied by yet another random number between 2 and 5. If the randomly generated value is less than or equal to 0.05. To facilitate the mutation process and avoid the ANN model from being trained in the wrong direction (resulting in a poorer training accuracy rate), some weighted matrix values can be slightly scaled. An ANN's performance could be impacted by a wide range of variables. The number of layers, the number of neurons in each layer, the learning rate, the optimization function, the loss function, and other factors are among them [43–47]. In a Genetic Algorithm, the ANN must be constructed while taking into account the population size, the number of generations, crossover rate, mutation rate, and probability. The model should provide a weighted matrix with a higher variance of values than those with a lower mutation rate and lower mutation probability, for example, if the mutation rate and its mutation probability are high. Its major objective is to attempt to address the flaw in the conventional gradient descent learning technique by ensuring that there is a variety of feasible weighted matrices rather than possibly training a model in an incorrect manner without obtaining the best possible solution for any problem.

### 5.2. PSO

PSO, which stands for Particle Swarm Optimization, is a metaheuristic optimization algorithm inspired by the behavior of birds flocking or fish schooling. It is commonly used in combination with artificial neural networks (ANNs) to optimize their performance. In the context of ANNs, PSO is employed to determine the optimal values of the ANN's parameters or weights. The algorithm operates by simulating a population of particles that move through a multidimensional search space. Each particle represents a potential solution or set of parameters for the ANN.

Initially, the particles are assigned random positions and velocities within the search space. The positions represent the values of the ANN's parameters, while the velocities determine the particles' movement direction and speed. The algorithm then evaluates the fitness or performance of each particle's solution by training the ANN with the corresponding parameter values and measuring its accuracy or error. During each iteration of the algorithm, the particles adjust their velocities and positions based on their own historical best solution (the best fitness achieved by the particle itself) and the global best solution (the best fitness achieved by any particle in the swarm). This collective learning and sharing of information guide the particles towards promising regions in the search space.

The adjustment of particle velocities is governed by two main factors: a cognitive component and a social component. The cognitive component represents the particle's own knowledge, which encourages it to move towards its best solution. The social component reflects the influence of the swarm and directs the particle towards the global best solution. As the iterations progress, the particles converge towards the optimal parameter values, which correspond to the weights that result in the best performance of the ANN.

The algorithm terminates when a specified stopping criterion is met, such as reaching a maximum number of iterations or achieving a desired level of performance [5,29,48–50]. By using PSO, ANN models can be effectively fine-tuned and optimized to improve their accuracy, convergence speed, and generalization capabilities. PSO provides a powerful approach for exploring and exploiting the parameter space of ANNs, leading to enhanced performance in a variety of tasks and applications.

### 5.3. BO-GP

Bayesian Optimization Gaussian Process (BO-GP) is a metaheuristic optimization algorithm commonly used for fine-tuning and optimizing artificial neural network (ANN) models. It combines Bayesian optimization, which is a sequential model-based optimization approach, with Gaussian processes, which are statistical models used for modeling the behavior of functions. In the context of ANN models, BO-GP aims to find the optimal set of hyperparameters that result in the best performance for the given task [51–53]. These hyperparameters include the number of hidden layers, the number of neurons in each layer, the learning rate, regularization parameters, activation functions, and other architectural choices. The BO-GP algorithm works iteratively as follows:

1. Define a search space: Determine the range or values for each hyperparameter that will be explored during the optimization process.
2. Create an initial design: Select a small set of initial hyperparameter configurations to evaluate the ANN model's performance. This initial design is often chosen using techniques like random sampling.
3. Build a surrogate model: Fit a Gaussian process regression model to the initial design data. The surrogate model approximates the behavior of the ANN model based on the observed hyperparameter performance pairs.
4. Optimize the acquisition function: The acquisition function guides the selection of the next set of hyperparameters to evaluate. It balances exploration (sampling new regions of the search space) and exploitation (focusing on promising regions). Common acquisition functions include Expected Improvement (EI), Upper Confidence Bound (UCB), and Probability of Improvement (PI).
5. Evaluate the ANN model: Select the next set of hyperparameters based on the optimized acquisition function and evaluate the performance of the ANN model with those hyperparameters. This involves training the ANN on a training set and evaluating its performance on a validation set or using cross-validation.
6. Update the surrogate model: Incorporate the new hyperparameter-performance pair into the existing data and update the Gaussian process regression model. This allows the surrogate model to improve its approximation of the ANN model's behavior.
7. Repeat steps 4 to 6: Iterate the process by optimizing the acquisition function, evaluating the ANN model, and updating the surrogate model until reaching a specified termination criterion (e.g., a maximum number of iterations or a desired level of performance).

The goal of BO-GP is to efficiently explore the hyperparameter search space and find the optimal configuration that maximizes the performance of the ANN model. By leveraging Bayesian optimization and Gaussian process modeling, BO-GP balances exploration and exploitation, enabling effective hyperparameter tuning and improving the overall performance of ANN models.

### 5.4. BO-TPE

Bayesian Optimization Tree-structured Parzen Estimator (BO-TPE) is another metaheuristic optimization algorithm commonly used for fine-tuning and optimizing artificial neural network (ANN) models. It is a variant of Bayesian optimization that employs a Tree-structured Parzen Estimator to model the performance of different hyperparameter configurations.

The BO-TPE algorithm works in the following steps:

1. Define a search space: Specify the range or values for each hyperparameter that will be explored during the optimization process.
2. Initialize the hyperparameter sampling: Randomly sample a set of hyperparameter configurations from the search space to create an initial design.
3. Evaluate the initial design: Train and evaluate the ANN models according to the initial set of hyperparameter configurations. This typically involves splitting the data into training and validation sets and using cross-validation or hold-out validation.
4. Build a probabilistic model: Based on the observed hyperparameter-performance pairs from the initial design, construct a probabilistic model using a Tree-structured Parzen Estimator. This model captures the relationship between hyperparameter values and the corresponding performance of the ANN models.
5. Update the model and sample new hyperparameters: The Tree-structured Parzen Estimator uses the previous observations and probabilistic model to guide the sampling of new hyperparameters for the next iteration. It balances exploration (sampling new regions) and exploitation (focusing on promising regions) by considering both the Probability of Improvement and the Expected Improvement.
6. Evaluate the new hyperparameter configurations: Train and evaluate the ANN models with the newly sampled hyperparameter configurations. Update the observed hyperparameter-performance pairs.
7. Update the probabilistic model: Incorporate the new observations into the probabilistic model. This allows the model to refine its estimation of the performance landscape and guide the sampling process more effectively.
8. Repeat steps 5 to 7: Iterate the process by sampling new hyperparameters, evaluating the ANN models, and updating the probabilistic model until reaching a termination criterion, such as a maximum number of iterations or a desired level of performance.

BO-TPE leverages the Bayesian optimization framework to efficiently explore the hyperparameter search space and identify the optimal configuration for the ANN model. By using a Tree-structured Parzen Estimator, it captures the complex relationship between hyperparameters and performance, enabling effective hyperparameter tuning and improving the overall performance of ANN models [54–56].

## 6. Results

Using the Keras model in Python, we evaluated ANN models trained using Adam (Adaptive Moment Estimation). The outputs of the various algorithms are shown below. It is evident from our results that both Bayesian optimization and metaheuristic algorithms, such as Particle Swarm Optimization (PSO) and Genetic Algorithm (GA), can both be effective for optimizing artificial neural networks (ANNs) due to their complementary strengths and capabilities. Bayesian optimization and metaheuristic algorithms can perform equally well for ANN optimization (Figure 12). Bayesian optimization is known for its ability to balance exploitation (exploiting known good solutions) and exploration (exploring new regions in the search space). It uses probabilistic models to guide the search towards promising regions, based on observed performance. On the other hand, metaheuristic algorithms like PSO and GA incorporate stochastic search techniques that can explore the search space more extensively, searching for global optima. ANNs often have a large number of hyperparameters, making the optimization problem high-dimensional. Bayesian optimization excels in handling high-dimensional spaces by building surrogate models that capture the relationship between hyperparameters and performance. It effectively guides the search to promising regions based on the surrogate model's predictions. Metaheuristic algorithms can also handle high-dimensional spaces through their population-based search, which allows for more diverse exploration.

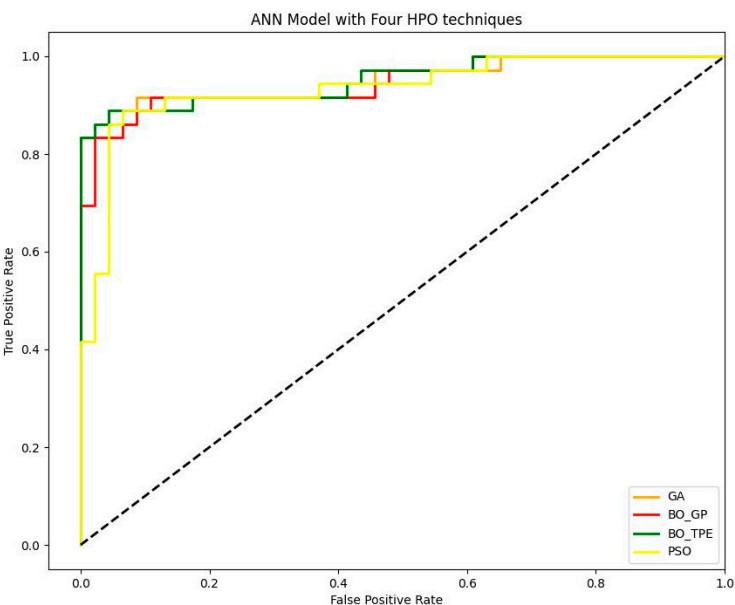

**Figure 12.** AUC value for GA, BO_GP, BO_TPE, and PSO for ANN model.

The performance landscape of ANNs can be complex and non-convex, with multiple local optima. Bayesian optimization and metaheuristic algorithms approach the optimization problem from different perspectives. Bayesian optimization focuses on modeling the performance landscape and exploiting its information to guide the search, while metaheuristic algorithms explore the landscape through heuristics and search operators to escape local optima. Both Bayesian optimization and metaheuristic algorithms offer flexibility in terms of the search strategy and problem representation. They can be applied to different types of optimization problems, including ANN optimization, by adapting their operators or objective functions accordingly. This adaptability allows them to be tuned and customized based on the specific requirements of the ANN optimization task.

Ultimately, the effectiveness of Bayesian optimization and metaheuristic algorithms for ANN optimization can depend on various factors, including the problem complexity, the size of the search space, and the availability of computational resources. It is often beneficial to experiment with different optimization techniques and select the one that performs best for a particular ANN optimization task.

The results of the AUC values (Table 12) indicate the performance of each HPO technique in optimizing the hyperparameters of the ANN for the given ML task. A higher AUC value generally indicates better discrimination power and predictive accuracy of the model.

**Table 12.** AUC value for GA, BO_GP, BO_TPE and PSO for ANN model.

| HPO Techniques | ML | AUC |
|:---:|:---:|:---:|
| GA | ANN | 0.950 |
| BO_GP | ANN | 0.947 |
| BO_TPE | ANN | 0.953 |
| PSO | ANN | 0.935 |

Based on the given data, BO_TPE achieved the highest AUC value of 0.95289, followed by GA with 0.94987, BO_GP with 0.94685, and PSO with 0.93538. This suggests that BO_TPE performed the best among the evaluated HPO techniques for the given ML task.

There could be several reasons why BO_TPE outperformed the other techniques. BO_TPE combines both exploration and exploitation strategies effectively. It intelligently

explores the hyperparameter space to identify promising regions and then exploits the information gained to refine the search for optimal hyperparameters. This balance between exploration and exploitation can lead to better performance. BO_TPE utilizes a sequential Bayesian optimization strategy, which leverages the information gathered from previous iterations to guide the search for optimal hyperparameters. This adaptive approach helps in efficiently exploring the hyperparameter space and quickly converging onto promising solutions. BO_TPE employs a Tree-structured Parzen Estimator (TPE) to efficiently explore the hyperparameter space. TPE focuses more on areas that are likely to contain better hyperparameter configurations, allowing for more efficient optimization compared to other techniques. BO_TPE utilizes the observed performance of previous hyperparameter configurations to build a probabilistic model and guide the search towards regions with higher potential. This exploitation of information helps in quickly identifying and refining good hyperparameter settings.

It is important to note that the performance of HPO techniques can vary depending on the specific dataset, ML task, and hyperparameter space being considered. The given results suggest that for the specific ML task and dataset in this study, BO_TPE performed better than GA, BO_GP, and PSO in optimizing the hyperparameters of the ANN model, resulting in higher predictive accuracy and discrimination power, as indicated by the AUC values.

The fact that GA and BO_TPE performed equally well, as shown in Table 12 and Figure 13, might suggest that both algorithms were able to navigate the search space effectively to find good solutions. As discussed earlier, GA uses a population-based approach, in which potential solutions evolve over generations through processes like mutation and crossover. BO_TPE, on the other hand, models the distribution of the objective function and uses Bayesian reasoning to focus the search on promising regions of the hyperparameter space. In this particular case, the characteristics of the problem may have allowed both techniques to reach similar levels of optimization performance.

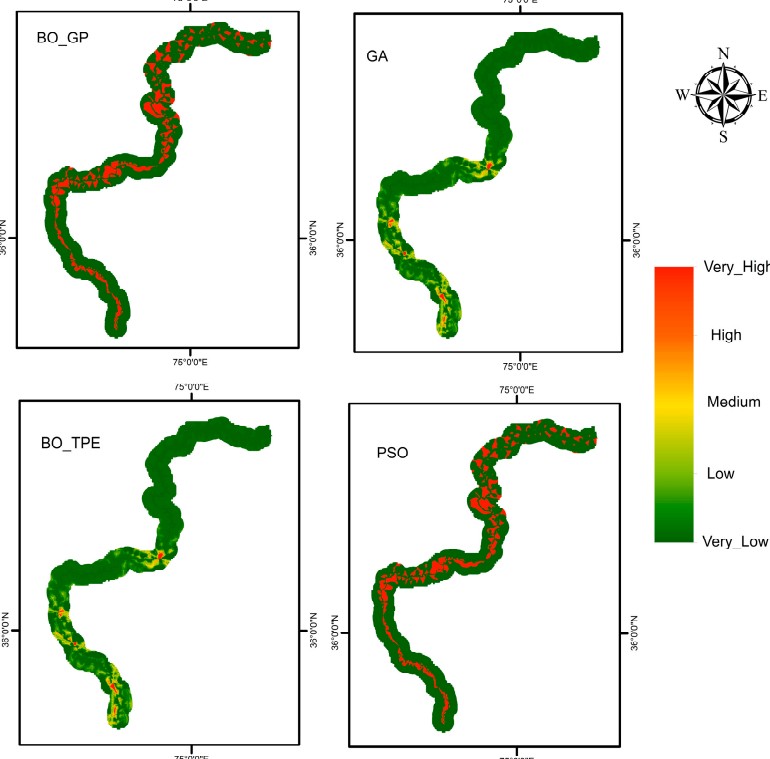

**Figure 13.** The landslide susceptibility maps generated by ANN using different optimization techniques, including BO_TPE, BO_GP, PSO and GA.

Bayesian Optimization Gaussian Processes (BO_GP) outperformed Particle Swarm Optimization (PSO) in terms of AUC. BO_GP's advantage might be attributed to its ability to model the objective function as a probabilistic surrogate, allowing it to make informed decisions about where to explore the hyperparameter space. PSO, while a powerful optimization technique inspired by social behavior of birds or fish, can sometimes struggle in complex or high-dimensional spaces due to its exploration and exploitation balance. It relies on particle movement guided by personal and global best solutions, which may not be as effective as BO_GP's probabilistic modeling for certain problems. It's possible that the optimization landscape of this problem favored the probabilistic modeling and informed exploration strategy employed by BO_GP, leading to its better performance compared to PSO.

## 7. Conclusions

In conclusion, this research endeavors to enhance the accuracy and reliability of predicting landslide susceptibility along the Karakoram Highway (KKH) by employing a comprehensive methodology that combines feature selection and hyperparameter optimization. This study acknowledges the intricate nature of landslide occurrences and the critical role that relevant geospatial variables and well-tuned model parameters play in accurate predictions. Through a rigorous process of experimentation and analysis, the study successfully integrates modern techniques to address the challenges posed by landslides in the Karakoram region. The utilization of artificial neural networks (ANNs) as predictive models, along with four distinct hyperparameter optimization techniques—Particle Swarm Optimization (PSO), Genetic Algorithm (GA), Bayesian Optimization Gaussian Process (BO_GP), and Bayesian Optimization Tree-structured Parzen Estimator (BO_TPE)—illustrates the commitment to refining model configurations for optimal performance. Furthermore, this study recognizes the significance of selecting pertinent features from the dataset. The incorporation of eight feature selection algorithms, each evaluating variables through different lenses, such as multicollinearity, correlations, information sharing, variance, and uncertainty, reflects a comprehensive approach to identifying the most influential geospatial variables contributing to landslide susceptibility. The synergy achieved by combining feature selection and hyperparameter optimization is instrumental in addressing the complexities of predicting landslide occurrences. The advantages encompass the prevention of overfitting, controlled model complexity, generalization to new and unseen data, the efficient utilization of computational resources, adaptation to varying contexts, and robustness in the face of changing data distributions. This approach is not only valuable for accurate predictions but also contributes to a deeper understanding of the underlying processes leading to landslides.

The study's findings provide a substantial contribution to the field of geospatial analysis, disaster management, and infrastructure planning. By offering insight into optimal feature selection and model parameterization, the research equips decision makers with valuable tools to develop effective mitigation strategies, plan resilient infrastructure, and manage hazards along the KKH. The combined methodology of feature selection and hyperparameter optimization stands as a testament to the potential of integrating advanced techniques to tackle complex challenges. The research not only enhances the accuracy of landslide susceptibility predictions but also underscores the broader applicability of such integrated approaches in various domains of data analysis and decision making. As natural calamities, like landslides, continue to pose threats to communities and infrastructure, this study exemplifies the kind of innovative thinking that can lead to more informed and proactive disaster management efforts.

**Author Contributions:** Conceptualization, F.A. (Farkhanda Abbas), F.A. (Fazila Abbas) and G.K.; Methodology, F.A. (Farkhanda Abbas) and F.A. (Fazila Abbas); Software, F.A. (Farkhanda Abbas); Validation, D.H.; Formal analysis, M.I., D.H. and M.F.A.; Investigation, M.I.; Resources, A.F.A. and M.F.A.; Data curation, M.I., J.I., G.K., A.F.A. and M.F.A.; Writing—original draft, F.A. (Farkhanda Abbas); Writing—review & editing, F.A. (Fazila Abbas) and J.I.; Supervision, F.Z.; Project administration, G.K. and A.F.A. All authors have read and agreed to the published version of the manuscript.

**Funding:** We extend our appreciation to the Researchers Supporting Project (no. RSP2023R218), King Saud University, Riyadh, Saudi Arabia.

**Data Availability Statement:** The data presented in the study are available upon request from the first and corresponding authors. The data are not publicly available due to the thesis that is being prepared using these data.

**Conflicts of Interest:** The authors declare no conflict of interest.

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
