# Peer review of "Landslide Susceptibility Mapping: Analysis of Different Feature Selection Techniques with Artificial Neural Network Tuned by Bayesian and Metaheuristic Algorithms"

_remotesensing, doi:10.3390/rs15174330_

Round 1

Reviewer 1 Report

  • The abstract is well-written and not in a professional manner. It provides an overview of the work carried out in this article.
  •  

·       Authors can add a paragraph related to the organization of the work done in different sections of article, as it will provides a short summary of work.

o   The aim or objective of the work must be clear and it would be better to write it in the introduction section.

o   The authors discussed about the water quality, efforts, action and strategies. But still fails to give a systematic model to improve these issues.

·       The conclusion must be written on basis of some comparison of previous techniques and this present technique in this manuscript.

  • The manuscript may be accepted after minor revision.

Author Response

kindly see attachment 

Reviewer 2 Report

I am pleased to have the opportunity to review an early version of this manuscript. The manuscript presents the use of metaheuristic and Bayesian techniques to train an artificial neural network (ANN) for landslide susceptibility mapping. Additionally, this manuscript explores eight feature selection algorithms to determine the importance of geospatial variables for landslide susceptibility mapping along the KKH. Although the paper is well-written and the experimental results are generally sound, there are several problems that need to be addressed. I suggest reconsidering this manuscript after a major revision.

Abstract: The abstract section provides a good introduction and overview of the full work but misrepresents the BO-TPE method used in the text on line 36. The BO-TPE is also misrepresented in several places throughout the paper.

Introduction: The introduction provides a comprehensive review of artificial neural networks, hyperparameter optimization algorithms, and feature selection algorithms used in this paper. While the introduction provides a good overview of the research progress and use of the algorithms, it does not sufficiently discuss the research progress related to landslide hazards. Please add more information on this topic into the introduction. See DOI: 10.1007/s12665-022-10723-z

Feature selection techniques: This section is an important part of the paper and outlines the principles of the feature selection techniques. However, the expressions of the formulas in this part are not clear enough and need modification. The explanation about the histogram is also too simple and could be expanded upon. Regarding the table of importance calculation on line 464, it lacks a detailed description of the calculation, so please provide a more detailed explanation. There is a conflict between the serial numbers of the figures in this section and the figures in the study area.

Results: The chapter lacks analysis and interpretation of the landslide sensitivity maps and does not analyze the four optimization algorithms used to characterize the differences in results. Additionally, the final landslide sensitivity maps are of different sizes, poorly ordered, and appear rough.

Minor points:

Line 80: There is a lack of figure numbering here, and it is recommended to check the full text for figure numbering issues.

Line 490: There is a problem with the figure name here, and it is recommended to change it.

Line 705: There is an issue with the numbering of the figure, and it is recommended to change it.

Line 770: The figures are of different sizes and poorly laid out. It is recommended to address this issue.

While I have raised the above issues, I think they need major revisions to the manuscript. I hope that my comments will help the authors prepare a more comprehensive final version of the paper.

Author Response

Kindly see attachment 

Reviewer 3 Report

The paper aims to map the landslide susceptibility maps in Karakoram highway by using ANN supervised by metaheuristic and Bayesian techniques (PSO, GA, BO_GP and BO_TPE). The geospatial factors used in the study were selected by 8 criteria - information gain, gain ratio, OneR classifier, subset evaluators, principal components, relief attribute evaluator, correlation, and symmetrical uncertainty. In terms of AUC, the BO_TPE has the best performance among the optimization algorithms in the study. The innovation of the paper could be further improved.

1.       There is no key word relevant to “landslide”.

2.       It is suggested to move the “Study area” section after the “Introduction” section.

3.       L34: Please modify the abbreviation "particle swarm optimization algorithm (PSO)" to "Particle Swarm Optimization (PSO) algorithm" to ensure consistent capitalization. Similarly, I request that you apply a similar pattern to other abbreviations throughout the document.

4.       L88 (and other similar text in the article): Please add a space between the end and start of a sentence, for example, “…Bayesian regularization [4].Some characteristics…” to “…Bayesian regularization [4]. Some characteristics, …”.

5.       L90, L105: When the corresponding abbreviation first appears in the text, the full name must be written. When the abbreviation first appears in the abstract, its full name needs to be written. When the abbreviation appears for the first time in the body of the text, the full name also needs to be written again using the form "full name (abbreviation)". For example, the “ANN” appears in L90 while its full name appears in L105, which should have appeared before L90. The author should check and revise the abbreviation problem throughout the document.

6.       L107: Please add a space between the citations and the word before brackets, for example, “Pakistan[6-9]” to “Pakistan [6-9]”.

7.       L140-L169: The 5 paragraphs in L140-L169 are relatively short and discuss the same topic, the optimization algorithms. It is suggested to merge them into one paragraph.

8.       L231-257: This paragraph is a conclusive description and is recommended to be placed after the factor selection process.

9.       L220-230: It is recommended to describe the existing problems in the research field and to summarize the experiment of the study in the last paragraph of the introduction.

10.    L232: What is "Road_dist"? It would greatly benefit the readers to explain the "Road_dist", or just write “the distance to the road”.

11.    L309: Why you choose these geospatial variables? Please add references.

12.    L314, Table 2: Please establish a consistent format of the first letter in every phrase. For example, the “Group” or “group” in “Chalt Group” and “Yasin group” should be consistent.

13.    L314, Table 2: Please remove the “m” in “0–100 m”. Change “000–1000 m” to “0–1000”.

14.    L262-265: Please revise this sentence: “As we discuss earlier that reducing the entropy and result into information loss and important patterns can be loss [24]therefore and we keep all the variable because the information gain for all of them was greater than zero.”

15.    Table 5 and Figure 3: Please establish the consistent number of decimal places. In Table 5 the accuracy score of “Road_dist” is 0.902439, while in Figure 3 it is 0.902439024390. Please check the whole text.

16.    There are two tables named “Table 2”, two tables named “Table 8”, and three figures named “Figure 1”. Please check the numbers.

17.    There are many wrong words in the text, please check the whole text. For example, “geospatail” should be “geospatial”.

18.    It is highly recommended to add numbers before the titles to make it easier for the reader to read it. For example, “1”, “1.1”, “1.2”, etc.

19.    L557-560, L707: It is not recommended to use "I" as a subject in article. You can use “We”.

20.    Please provide a more in-depth discussion of the limitations of this study based on relevant studies.

21.    L770: Please move north arrow and legend in the right side of the maps, which would be more reader-friendly.

22.    L747: What is HPO? Please add the full name - Hyperparameter Optimization (HPO) - for the first time appears in the text. Similar for “AUC”. Please check other abbreviations.

23.    There is no description or discussion of the landslide susceptibility maps, please add it.

24.    There is no description of the datasets used in the study, please add it.

This manuscript must be improved by a native English speaker.

Author Response

kindly see attachment. 

Round 2

Reviewer 2 Report

There are some concerns that need to be addressed:

1. Table Formatting: There is an issue with the formatting of the first row in the contents of Table 2. This formatting error should be rectified.

2. Methodology Chapter: The methodology chapter appears to be too short and lacks sufficient content. I recommend expanding upon the methodology section to provide more comprehensive details.

3. Formula Formatting Errors: There are several formatting errors within the manuscript, particularly in the formula part and numbered paragraph settings. These formatting errors should be corrected to maintain a professional appearance.

4. Histograms: In Figure 4 and similar histograms, the sizes of the histograms and axes font sizes should be made consistent for improved visual presentation.

Overall, I believe that addressing these concerns would require revisions. I hope that my comments will assist the authors in preparing a more refined and comprehensive final version of the paper.

Author Response

kindly s

Reviewer 3 Report

The authors have revised the paper accordingly, but there is no change about “L770: Please move north arrow and legend in the right side of the maps, which would be more reader-friendly.” -Please modify the Figure 13 (L1195). And please add the discussion and conclusions.

Minor editing of English language required.

Author Response

kindly see attachment 
